# PlanarGS: High-Fidelity Indoor 3D Gaussian Splatting Guided by Vision-Language Planar Priors

**Xirui Jin**[*]
Shanghai Jiao Tong University

**Renbiao Jin**[*]
Shanghai Jiao Tong University

**Boying Li** [†]
Monash University
boying.li@monash.edu

**Danping Zou**[†]
Shanghai Jiao Tong University
dpzou@sjtu.edu.cn

**Wenxian Yu**
Shanghai Jiao Tong University

## Abstract

Three-dimensional Gaussian Splatting (3DGS) has recently emerged as an efficient representation for novel-view synthesis, achieving impressive visual quality. However, in scenes dominated by large and low-texture regions, common in indoor environments, the photometric loss used to optimize 3DGS yields ambiguous geometry and fails to recover high-fidelity 3D surfaces. To overcome this limitation, we introduce PlanarGS, a 3DGS-based framework tailored for indoor scene reconstruction. Specifically, we design a pipeline for Language-Prompted Planar Priors (LP3) that employs a pretrained vision-language segmentation model and refines its region proposals via cross-view fusion and inspection with geometric priors. 3D Gaussians in our framework are optimized with two additional terms: a planar prior supervision term that enforces planar consistency, and a geometric prior supervision term that steers the Gaussians toward the depth and normal cues. We have conducted extensive experiments on standard indoor benchmarks. The results show that PlanarGS reconstructs accurate and detailed 3D surfaces, consistently outperforming state-of-the-art methods by a large margin. Project page: https://planargs.github.io

## 1 Introduction

Image-based 3D modeling is a long-standing challenge in the computer vision community, aiming to recover the 3D scene from a collection of images [42]. It serves as a fundamental component for a wide range of applications, including computer graphics, augmented/virtual reality (AR/VR), and robotics. Recently, 3D Gaussian Splatting (3DGS) [18] has emerged as a promising approach to modeling the 3D world. Given pictures captured from different viewpoints, 3DGS optimizes both the appearance and the geometry (scale, rotation, and position) of 3D Gaussian spheres to enforce the rendered image to be as close as possible to the real image captured in the target viewpoint. The appearance similarity, described by a photometric loss, plays in key role in the 3DGS training process.

Despite its impressive visual quality of view rendering, 3DGS still struggles to reconstruct a high-fidelity 3D surface model in scenes where large low-texture regions exist. Such kind of scenes are commonly seen in indoor environments. Lack of textures makes using the appearance cue as the supervision signal alone difficult to resolve the geometric ambiguities, resulting in inaccurate or false geometric reconstruction. To resolve such ambiguity, the recent method PGSR [4] applies the multi-view geometric consistency constraint to enhance the smoothness in areas with weak textures.

---

[*]Equal contribution.

[†]Corresponding author.

39th Conference on Neural Information Processing Systems (NeurIPS 2025).

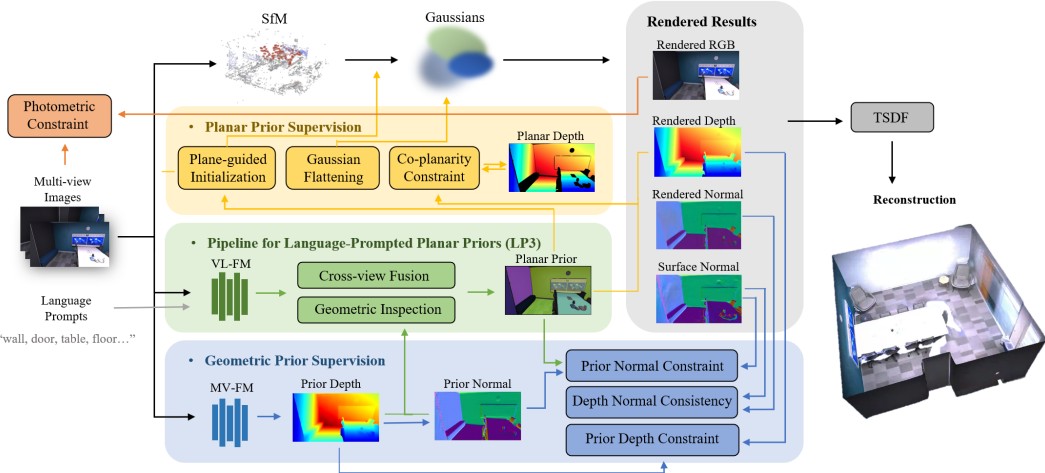

Figure 1: **PlanarGS overview.** Our method takes multi-view images and language prompts as inputs, getting planar priors through the pipeline for Language-Prompted Planar Priors (LP3). Our planar prior supervision includes plane-guided initialization, Gaussian flattening, and the co-planarity constraint, accompanied by geometric prior supervision. Both foundation models in the figure are pretrained.

Other approaches [6, 46, 12] have incorporated monocular depth or normal priors in the optimization process. However, existing methods still cannot handle large and texture-less planar regions. Those methods tend to produce locally smooth and globally curvy surfaces instead of perfect planes for those planar regions. One feasible solution is to explicitly detect those planar regions and enforce the planar geometry on these regions. This idea has been explored in monocular depth estimation [22, 61, 36] and Neural Radiance Field (NeRF) [13]. Those methods, however, highly rely on the performance of a heuristic plane detector [22] or a standalone neural plane detector [13] trained from limited data. However, such kinds of plane detectors often produce noisy segmentation and are difficult to apply to novel scenes (e.g, unseen planar objects not labeled in the training dataset).

In this paper, we introduce a novel framework of 3D Gaussian Splatting, **PlanarGS**, which leverages pretrained foundation models to overcome the aforementioned challenges of indoor 3D reconstruction. Foundation models, typically trained on vast, diverse datasets, exhibit far greater generalization and better performance than smaller, task-specific networks. To detect the planar regions in the image, we employ a pretrained vision-language segmentation model [39] prompted with terms such as "wall", "door", and "floor". Using word prompts enables flexible adaptation to new scenes by simply swapping or adding keywords, so the model can identify novel planar surfaces. For instance, in the classroom, one might include the extra word "blackboard" as the prompt to capture the novel planar structures not found in a home environment.

The detection results by the vision-language segmentation model always contain misleading region proposals, so we design the pipeline for Language-Prompt Planar Priors (**LP3**) to obtain reliable planar priors. For example, it is common that two connected perpendicular walls are merged into a single planar region. To resolve such ambiguities, additional geometric cues are necessary. Moreover, per-image detection frequently omits certain planes because it relies solely on local image information. Incorporating multi-view context can recover these missing labels by aggregating evidence from adjacent frames. Accordingly, we employ DUSt3R [50], a pretrained multi-view 3D foundation model, to extract the depth and normal maps from neighboring images. Those maps serve as both the extra cues for more accurate planar region identification and as the geometric priors to guide the 3DGS optimization. We opt for DUSt3R's multi-view approach rather than a monocular alternative such as [56, 57] because it produces view-consistent 3D structures, enabling robust fusion of region proposals across views and yielding stronger 3D priors for the optimization process.

To incorporate planar priors into detected planar regions, we introduce a planar supervision scheme during 3DGS optimization. This includes plane-guided initialization, Gaussian flattening, and a co-planarity constraint, which together help preserve global surface flatness and suppress local artifacts.

Depth and normal priors from the pretrained 3D model are applied to under-constrained regions. We impose a prior normal constraint for planar regions, encouraging the surface normals extracted from 3D Gaussians to align with the per-pixel normals predicted by DUSt3R. For low-texture regions, we constrain the rendered depth to match the scale-and-shift corrected prior depth and enforce consistency between the predicted depth and normal. Finally, to compensate for insufficient SfM points in texture-less planes, we densely initialize Gaussians by randomly sampling pixels within each detected plane and using their 3D coordinates, which are back-projected from the depth priors, as initial Gaussian centers.

Our main contributions include:

- We introduce a novel 3DGS framework that integrates semantic planar priors and multi-view depth priors to achieve high-fidelity reconstruction in challenging indoor scenes.

- We design a robust pipeline for Language-Prompted Planar Priors (LP3) that applies vision-language foundation models with cross-view fusion and geometric inspection to obtain reliable planar priors.

- Our method achieves state-of-the-art performance on high-fidelity surface reconstruction tasks across the Replica [44], ScanNet++ [60], and MuSHRoom [40] datasets.

## 2 Related Work

**Image-based 3D Modeling:** Image-based 3D modeling has been a long-standing goal in computer vision. The typical pipeline is as follows. Firstly, structure-from-motion (SfM) [42] is used to recover camera poses and sparse 3D points by matching feature points across different views. Then, based on the posed images, Multi-view stereo (MVS) [2, 43, 58, 32] generates dense depth maps from multi-view images via dense image matching, which are then converted to dense point clouds. Finally, surface reconstruction [34, 17] is applied to recover the 3D mesh and its texture from the point clouds, yielding the final 3D representation of the scene. This pipeline, successfully applied to outdoor photogrammetry for decades, still struggles to reconstruct accurate 3D models in texture-less scenes, such as indoor environments that usually contain large, texture-less planar regions. Moreover, using textured meshes for 3D scene representation often exhibits holes or other artifacts when rendered from novel viewpoints. Recently, implicit and point-based representations such as Neural Radiance Fields (NeRF) and Gaussian Splatting have emerged, offering superior expressiveness for jointly modeling geometry and appearance.

**Neural Radiance Field (NeRF):** Neural Radiance Field (NeRF) [33] utilizes a multilayer perceptron (MLP) to model a 3D scene as a continuous 5D radiance field optimized via volumetric rendering of posed images. However, because NeRF's training also relies on photometric cues, it still struggles to recover accurate geometry in texture-less regions. To address this limitation, existing approaches have explored incorporating various priors. Many of them [48, 62, 45, 9, 35] use data-driven monocular depth or normal prediction as priors to improve the quality of the results. Other approaches incorporate indoor structural cues into reconstruction. For example, Manhattan-SDF [13] enforces a Manhattan-world assumption [11] by constraining floors and walls to axis-aligned planes, and another method [70] relaxes the constraints to the Atlantaworld assumption. Although better results have been achieved, their methods depend on a standalone plane detector trained on limited datasets, which reduces the flexibility across diverse scenes. Moreover, Manhattan-SDF requires an additional MLP to fuse noisy plane segmentations across views by comparing predicted labels from this MLP to observations. It is, however, not easy to extend such MLP fusion strategies to Gaussian Splatting, which represents scenes purely with spherical Gaussians.

**Gaussian Splatting:** Comparing with NeRF, 3D Gaussian Splatting [18] represents scenes explicitly with 3D Gaussians, which is more flexible and interpretable, as well as achieves significantly faster training and rendering speed. Similar to NeRF, 3DGS also depends on photometric loss for training. It therefore faces the same challenge in the texture-less environments. To enhance geometry reconstruction in Gaussian Splatting, numerous approaches [26, 46, 6, 12] also utilize monocular depth or normal priors for optimization. PGSR [4] introduces multi-view consistency, and FDS [7] leverages optical flow priors, both demonstrating performance gains. In addition, structure cues of indoor scenes have been exploited to guide Gaussian shapes and densification [1, 66, 41, 27]. Some methods jointly optimize Gaussian reconstruction with other representations, which are effective but

cumbersome. For example, GaussianRoom [53] combines with neural SDF [49], while other works incorporate textured mesh [16] or semantic maps [52, 38].

However, the aforementioned methods either focus on constraints of local regions or lack effective optimization mechanisms, making it difficult to reconstruct globally smooth surfaces. Our method designs a pipeline that adapts foundation models to obtain accurate planar priors, allowing for the holistic enforcement of co-planarity constraints in 3DGS. Meanwhile, multi-view depth priors also provide geometric scale information for PlanarGS.

## 3 Method

The whole pipeline of PlanarGS is illustrated in Fig. 1. Firstly, we apply a pretrained multi-view 3D foundation model [50] to obtain depth and normal priors. Subsequently, We leverage a pretrained vision-language foundation model [39] to the input images. Planar prior masks are generated from the model's outputs through our LP3 pipeline (Sec. 3.1 **Pipeline for Language-Prompted Planar Priors**), which implements cross-view fusion and geometric verification using the depth and normal priors. In addition, the depth and normal priors serve to guide 3DGS optimization (Sec. 3.4 **Geometric Prior Supervision**). Notably, we propose a planar supervision mechanism (Sec. 3.3 **Planar Prior Supervision**) that flattens Gaussians and utilizes the planar priors to guide initialization and enforce the co-planarity constraint during optimization. Finally, the high-fidelity 3D mesh can be recovered from the rendered depth maps from PlanarGS via TSDF fusion [31, 34].

### 3.1 Preliminary: 3D Gaussian Splatting

Three-Dimensional Gaussian Splatting [18] represents scenes with a set of Gaussians $\{G_i\}$ initialized by the sparse SfM result. Each Gaussian is defined by the center $\boldsymbol{\mu}_i$ and a full 3D covariance matrix $\boldsymbol{\Sigma}_i$ that can be factorized into a scaling matrix $\boldsymbol{S}_i$ and a rotation matrix $\boldsymbol{R}_i$:

$$G_i(\boldsymbol{x}) = e^{-\frac{1}{2}(\boldsymbol{x}-\boldsymbol{\mu}_i)^T \boldsymbol{\Sigma}_i^{-1}(\boldsymbol{x}-\boldsymbol{\mu}_i)} \qquad \boldsymbol{\Sigma}_i = \boldsymbol{R}_i \boldsymbol{S}_i \boldsymbol{S}_i^T \boldsymbol{R}_i^T. \tag{1}$$

To project 3D Gaussians to 2D for rendering, the center and the covariance matrix of Gaussian $G_i$ can be projected to 2D coordinates as [71]:

$$\boldsymbol{\mu}_i' = \boldsymbol{K}\boldsymbol{W}[\boldsymbol{\mu}_i, 1]^T \qquad \boldsymbol{\Sigma}_i' = \boldsymbol{J}\boldsymbol{W}\boldsymbol{\Sigma}_i \boldsymbol{W}^T \boldsymbol{J}^T, \tag{2}$$

where $\boldsymbol{W}$ is the viewing transformation matrix and $\boldsymbol{J}$ is the Jacobian of the affine approximation of the projective transformation. Rendering color $\boldsymbol{C}$ of a pixel can be given by $\alpha$-blending [20]:

$$\hat{\boldsymbol{C}} = \sum_{i \in N} T_i \alpha_i \boldsymbol{c}_i \qquad T_i = \prod_{j=i}^{i-1}(1 - \alpha_j), \tag{3}$$

where $\boldsymbol{c}_i$ and $\alpha_i$ represents color and density of the Gaussian and $N$ is the number of Gaussians the ray passes through.

### 3.2 Pipeline for Language-Prompted Planar Priors

Existing plane segmentation methods [28, 55, 68, 65, 54, 51, 29] primarily design small and dedicated networks, which rely heavily on extensive annotated datasets and suffer from inaccurate segmentation. While they can be applied for coarse plane extraction and reconstruction, their results cannot serve as planar priors that impose strong constraints for high-quality 3D reconstruction, as ZeroPlane [29] in Fig. 5. By comparison, our pipeline for Language-Prompted Planar Priors (abbreviated as **LP3**) employs a vision-language foundation model GroundedSAM [39], applying cross-view fusion and geometric verification to its segmentation outputs to generate planar priors, as in Fig. 2. Simultaneously, since 3DGS is trained specifically for every scene, the tunability of prompts allows PlanarGS to achieve better reconstruction in atypical scenes.

**Cross-view Fusion:** We first provide text prompts to the object detection foundation model [30] and get bounding boxes. However, we found that large planes often go undetected in single images because their regions either span beyond the frame or lie close to the image boundaries. As shown in Fig. 2(a), to address the plane omission problem, we utilize the multi-view consistency from neighboring viewpoints for the cross-view fusion of plane boxes. Specifically, we leverage prior depth maps $D_r$ (see Sec. 3.4) to back-project the main plane mask pixels $\boldsymbol{p}_s$ from a neighboring frame (source frame) into 3D points $\boldsymbol{P}_s$:

$$\boldsymbol{P}_s = D_r(\boldsymbol{p}_s) \cdot \boldsymbol{K}^{-1}\widetilde{\boldsymbol{p}}_s, \tag{4}$$

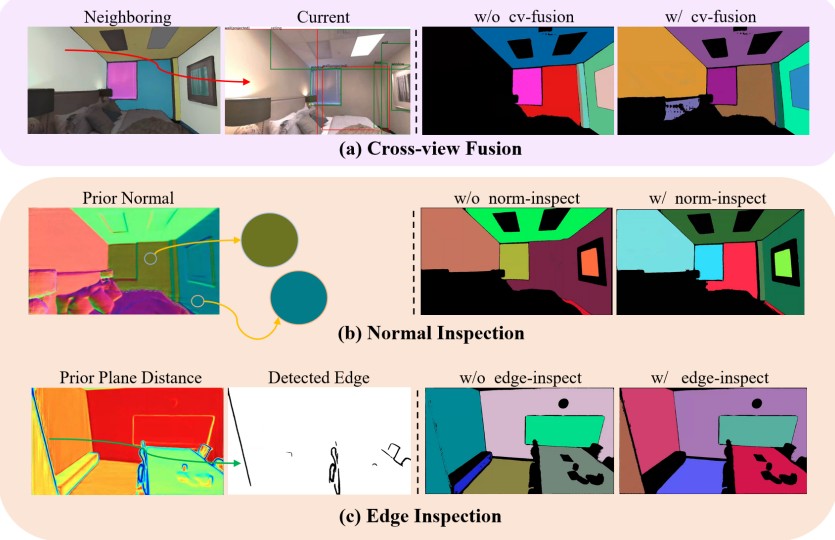

Figure 2: **Pipeline for Language-Prompted Planar Priors (LP3).** (a) We employ cross-view fusion to supplement bounding box proposals. (b)(c) Prior normal and plane-distance maps are incorporated for geometric inspection. Consequently, we obtain abundant and accurate planar priors from this robust pipeline.

where $\boldsymbol{K}$ is the intrinsic matrix, $\boldsymbol{P}_s = [x, y, z]^T$ is in the camera coordinate system of source frame, $\widetilde{\boldsymbol{p}}_s = [u, v, 1]^T$ is the homogeneous coordinate of the normalized pixel $\boldsymbol{p}_s$ in the source camera. $\boldsymbol{P}_s$ is then transformed into the camera coordinate system of the current frame (target frame) $\boldsymbol{P}_t$ and finally re-projected to 2D pixels $\boldsymbol{p}_t$:

$$\boldsymbol{P}_t = \boldsymbol{R}_t \boldsymbol{R}_s^T \boldsymbol{P}_s + (\boldsymbol{t}_t - \boldsymbol{R}_t \boldsymbol{R}_s^T \boldsymbol{t}_s) \qquad z_t \widetilde{\boldsymbol{p}}_t = \boldsymbol{K}_t \boldsymbol{P}_t. \tag{5}$$

After filtering out nested boxes with the same labels, we obtain refined detection results for downstream segmentation.

**Geometric Inspection:** Subsequently, we generate pixel-scale segmentation masks through the foundation model [19] according to the detected boxes. Nevertheless, sometimes the segmentation masks may incorrectly cover multiple planes together due to imprecise boxes, so we employ inspection from geometric priors as shown in Fig. 2(b)(c). We first convert depth priors $D_r$ into surface-normal priors $\boldsymbol{N}_{dr}$ using the local plane assumption [37, 4]. For each pixel $\boldsymbol{p}$ in the depth map, we sample its four neighboring pixels and project these pixels into 3D points as Eq. 4, then the surface normal of the local plane at pixel $\boldsymbol{p}$ is:

$$\boldsymbol{N}_{dr}(\boldsymbol{p}) = \frac{(\boldsymbol{P_1} - \boldsymbol{P_0}) \times (\boldsymbol{P_3} - \boldsymbol{P_2})}{|(\boldsymbol{P_1} - \boldsymbol{P_0}) \times (\boldsymbol{P_3} - \boldsymbol{P_2})|}. \tag{6}$$

We then express the distance from the local plane to the camera as $\delta_r(\boldsymbol{p}) = \boldsymbol{P} \cdot \boldsymbol{N}_{dr}(\boldsymbol{p})$ and get a prior plane-distance map $\delta_r$. We apply the K-means clustering to partition the prior normal map $\boldsymbol{N}_{dr}$ to separate non-parallel planes. For the separation of parallel planes, we identify the outliers of the plane-distance map as geometric edges that violate the local plane assumption. This stage also splits non-planar areas into fragments so we can filter them out.

### 3.3   Planar Prior Supervision

Inspired by previous works [61, 22, 21, 23], we leverage semantic planar priors as a holistic supervision for indoor reconstruction. Unlike existing approaches [13, 70, 8, 59, 64] that incorporate semantic features, our method treats the planar prior generation as a pre-processing stage (see Sec. 3.2), offering a simpler solution.

**Plane-guided Initialization:** The Gaussian initialization based on SfM relies on feature extraction, leading to point cloud deficiency in low-texture regions. We back-project pixels of plane areas to dense 3D points $\boldsymbol{P}_m$ as Eq. 4, while obtaining the plane labels of existing Gaussian points by

projecting them to 2D. For each plane region in each viewpoint, we calculate the average distance of existing Gaussians, deciding whether to supplement it with dense 3D plane points $\boldsymbol{P}_m$.

**Gaussian Flattening and Depth Rendering:**   In geometric reconstruction, flattening the Gaussians into planes enables them to conform more closely to the real-world surfaces, meanwhile producing less ambiguous depth and normal renderings [4]. Accordingly, we minimize the minimum scale factor $\boldsymbol{S}_i = diag(s_i, s_2, s_3)$ for each Gaussian [5] and define the minimum scale factor as the normal of the Gaussian $\boldsymbol{n}_i$. The normal map of the current viewpoint can be rendered through $\alpha$-blending with the rotation matrix from the camera to the global world $\boldsymbol{R}_c$:

$$L_s = ||min(s_1, s_2, s_3)||_1 \qquad \hat{\boldsymbol{N}} = \sum_{i \in N} \boldsymbol{R}_c^T \boldsymbol{n}_i \alpha_i \prod_{j=i}^{i-1}(1 - \alpha_j). \tag{7}$$

For each flattened Gaussian, the plane distance $d_i$ to the camera center $\boldsymbol{O}_c$ can be expressed by projecting the distance of the Gaussian center $\boldsymbol{\mu}_i$ to the normal direction $\boldsymbol{n}_i$. The rendered plane-distance map $\hat{\delta}$ is also obtained through $\alpha$-blending:

$$d_i = (\boldsymbol{R}_c^T(\boldsymbol{\mu}_i - \boldsymbol{O}_c))\boldsymbol{R}_c^T \boldsymbol{n}_i^T \qquad \hat{\delta} = \sum_{i \in N} d_i \alpha_i \prod_{j=i}^{i-1}(1 - \alpha_j). \tag{8}$$

Following method [4], we obtain the rendered depth map $\hat{D}$ of Gaussians from the rendered plane-distance $\hat{\delta}$ and normal map $\hat{\boldsymbol{N}}$:

$$\hat{D}(\boldsymbol{p}) = \frac{\hat{\delta}(\boldsymbol{p})}{\hat{\boldsymbol{N}}(\boldsymbol{p})\boldsymbol{K}^{-1}\widetilde{\boldsymbol{p}}}, \tag{9}$$

where $\boldsymbol{p} = [u, v]^T$ is the pixel position and $\widetilde{\boldsymbol{p}}$ is the homogeneous coordinate of $\boldsymbol{p}$.

**Co-planarity Constraint:**   To incorporate the holistic planar priors into Gaussian optimization, we apply the co-planarity constraint on the depth map. We first back-project the rendered depth map $\hat{D}$ to dense 3D points as Eq. 4. Following methods [61, 22], we define the plane $p_m$ in the 3D space as $\boldsymbol{A}_m^T \boldsymbol{P} = 1$, where $\boldsymbol{A}_m$ is the plane parameter and its direction represents the normal of the plane. As a result, the dense 3D points within the plane region $p_m$ should follow an equation $\boldsymbol{Q}_n \boldsymbol{A}_m = \boldsymbol{Y}_m$ and $\boldsymbol{A}_m$ can be obtained by solving the least squares problem:

$$\boldsymbol{A}_m = (\boldsymbol{Q}_n^T \boldsymbol{Q}_n + \epsilon \boldsymbol{E})^{-1} \boldsymbol{Q}_n^T \boldsymbol{Y}_m, \tag{10}$$

where $\boldsymbol{Q}_n = [\boldsymbol{P}_1, \boldsymbol{P}_2...\boldsymbol{P}_n]^T$, $\boldsymbol{Y}_n = [1, 1...1]^T$ and $\epsilon$ is a small scale to ensure numerical stability with an identity matrix $\boldsymbol{E}$. Subsequently, we obtain the planar depth map $D_p$ as:

$$D_p(\boldsymbol{p}) = (\boldsymbol{A}_m^T \boldsymbol{K}^{-1}\widetilde{\boldsymbol{p}})^{-1}. \tag{11}$$

The depth $D_p$ obtained from plane fitting is then used as a co-planarity constraint to the rendered depth. The loss function is defined as:

$$L_p = \frac{1}{N_p} \sum_{\boldsymbol{p} \in p} ||D_p(\boldsymbol{p}) - \hat{D}(\boldsymbol{p})||_1, \tag{12}$$

where $p$ represents all plane regions and $N_p$ represents the number of pixels in plane regions. Curved objects and clutter in the scenes are effectively excluded by our planar masks. Since these objects usually exhibit rich textures and geometric details, their reconstruction is less problematic.

### 3.4   Geometric Prior Supervision

Previous methods mostly introduce depth priors from monocular depth estimation, which suffer from local misalignments [67, 14, 24, 25]. We instead utilize the pretrained multi-view foundation model [50] and provide view-consistent geometric supervision of Gaussian optimization together with the planar priors.

**Prior Depth Constraint:**   We align the resized dense depth map $D_{dense}$ obtained from DUSt3R with the sparse depth $D_{sparse}$ derived by projecting the SfM-reconstructed point cloud to the current viewpoint. Because of the memory consumption, the depth maps are generated in groups, and each group shares the same scale and shift parameters.

$$s^*, t^* = \underset{s,t}{\arg\min} \sum_{\boldsymbol{p} \in D_{sparse}} ||D_{sparse}(\boldsymbol{p}) - (s \cdot D_{dense}(\boldsymbol{p}) + t)||_1, \tag{13}$$

and we utilize the aligned dense depth map as depth prior: $D_r = s^* \cdot D_{dense} + t^*$. We leverage the prior depth as a constraint on the rendered depth:

$$L_{rd} = \frac{1}{N_{lt}} \sum_{\boldsymbol{p} \in lt} M_{cof}(\boldsymbol{p}) \cdot ||D_r(\boldsymbol{p}) - \hat{D}(\boldsymbol{p})||^2, \tag{14}$$

where $M_{cof}$ is the mask from the confidence map of DUSt3R for its depth prediction. The $lt$ represents the collection of pixels in low-texture regions, generated by computing a mask through the Canny edge detector [3] on RGB input to mitigate the effect of depth priors' low resolution.

**Prior Normal Constraint:** The prior depth constraint serves as a scale-based supervision, while normal supervision offers greater flexibility in certain scenes and has been widely adopted in recent studies [4, 6, 12]. Following Eq. 4 and Eq. 6, we can obtain the surface-normal map $\hat{N}_d$ from the rendered depth $\hat{D}$. We constrain the rendered surface-normal $\hat{N}_d$ with the prior surface-normal $N_{dr}$:

$$L_{rn} = ||N_{dr} - \hat{N}_d||_1 + (1 - N_{dr} \cdot \hat{N}_d) \quad (\boldsymbol{p} \in p). \tag{15}$$

The normal prior is exclusively utilized on the plane region $p$ we extracted, optimizing the position of Gaussians together with the co-planarity constraint in Sec. 3.3.

**Depth Normal Consistency:** Additionally, inspired by works [4, 6, 12], we introduce the depth normal consistent regularization between rendered GS-normal $\hat{N}$ and rendered surface-normal $\hat{N}_d$ to optimize the rotation of Gaussians consistent with their positions:

$$L_{dn} = \frac{1}{N_{lt}} \sum_{\boldsymbol{p} \in lt} \cdot ||\hat{N}_d(\boldsymbol{p}) - \hat{N}(\boldsymbol{p})||_1, \tag{16}$$

where lt is the same as Eq. 14 to avoid edge areas that violate the local plane assumption in Eq. 6.

### 3.5 Optimization

The overall loss function of PlanarGS that combines planar and geometric supervision is:

$$L_{total} = L_{RGB} + L_s + \lambda_1 L_{dn} + \lambda_2 L_p + \lambda_3 L_{rd} + \lambda_4 L_{rn}, \tag{17}$$

where $\lambda$ are parameters and $L_{RGB}$ includes $L_1$ and D-SSIM loss as 3DGS.

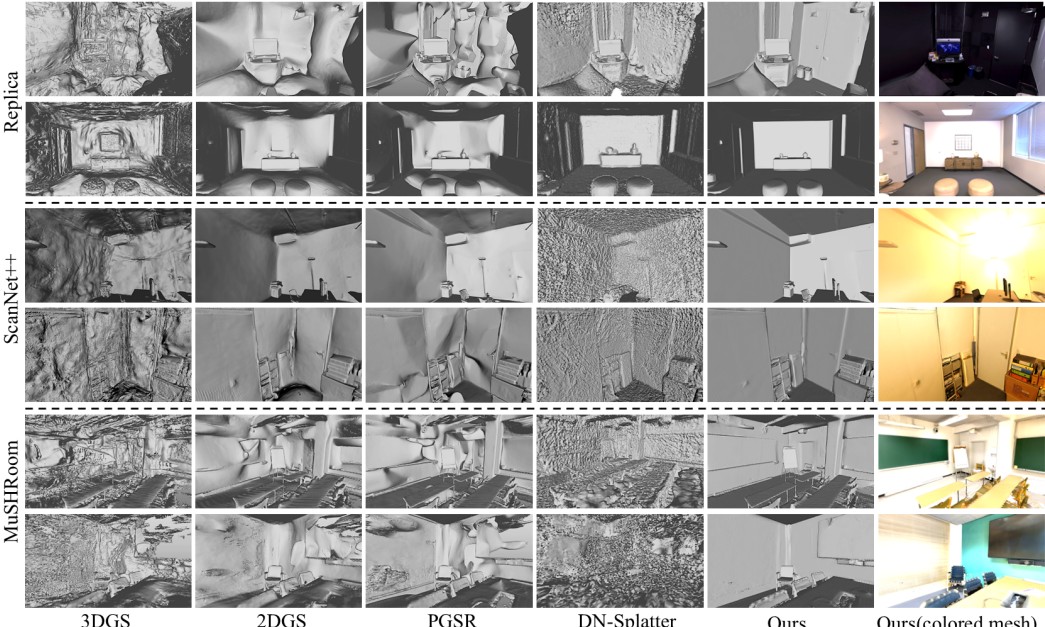

Figure 3: **Qualitative comparison.** We present reconstructed meshes from other methods and PlanarGS. The right column shows our colored meshes. The results demonstrate that PlanarGS achieves more accurate and comprehensive high-fidelity mesh reconstruction.

# 4 Experiments

We demonstrate the superior performance of PlanarGS on 3D reconstruction of indoor scenes in Sec. 4.1 and conduct ablation studies in Sec. 4.2.

**Datasets:** Following other works [41, 46] on Gaussian reconstruction in indoor scenes, we conduct our experimental evaluation on three indoor scene datasets, including 8 scenes from the synthetic dataset Replica [44] and 9 scenes from the real-world datasets: 4 scenes from ScanNet++ [60] and 5 complex scenes from MuSHRoom [40]. We use COLMAP [42] for all these scenes to generate a sparse point cloud as initialization.

**Implementation Details:** All experiments in this paper are conducted on the Nvidia RTX 3090 GPU. The training iterations for all scenes are set to 30,000, and $\lambda_1$, $\lambda_2$, $\lambda_3$, $\lambda_4$ are set as 0.05, 0.5, 0.05, 0.2, respectively. We use "wall, floor, door, screen, window, ceiling, table" as prompts for planar prior generation on Replica and ScanNet++ datasets, as these classes are commonly found in various indoor environments, and additionally add "blackboard, sofa" on the MuSHRoom dataset.

**Evaluation Metrics:** Consistent with existing methods [46, 7, 38] to evaluate the quality of reconstructed surfaces, we report Accuracy (Acc), Completion (Comp), and Chamfer Distance (CD) in cm. F-score (F1) with a 5cm threshold and Normal Consistency (NC) are reported as percentages. For novel view synthesis (NVS), we follow standard PSNR, SSIM, and LPIPS metrics for rendered images.

## 4.1 Reconstruction Results

We conduct comparisons with the baseline 3DGS [18], and Gaussian surface reconstruction methods 2DGS [15], GOF [63], PGSR [4], QGS [69], DN-Splatter [46]. For 3DGS, we obtain depth maps by applying alpha-blending integration to the z-coordinates of the Gaussians and generate a mesh through TSDF. For DN-Splatter, we apply depth priors from the multi-view foundation model that same as ours.

**Quantitative Results:** The quantitative comparison on three datasets are presented in Tab. 1 and Tab. 2. The reconstruction results of our method significantly outperform those of other methods across benchmarks, while simultaneously ensuring high-quality novel view synthesis in Tab. 1. Additionally, PlanarGS requires training time within one hour, comparable to other 3DGS-based methods. We also generate meshes directly from the depth priors provided by DUSt3R using TSDF, as shown in Tab. 2, to demonstrate the high-fidelity advantage of Gaussian splatting methods.

Table 1: Quantitative results of surface reconstruction and NVS on **MuSHRoom** dataset. Our method outperforms other methods on most metrics. The best results are marked in **bold**.

|  | Surface Reconstruction | | | | | NVS | | |
|---|---|---|---|---|---|---|---|---|
|  | Acc↓ | Comp↓ | CD↓ | F1↑ | NC↑ | PSNR↑ | SSIM↑ | LPIPS↓ |
| 3DGS [18] | 12.01 | 11.85 | 11.92 | 38.53 | 62.00 | 25.79 | 0.8775 | 0.2059 |
| 2DGS [15] | 9.16 | 10.27 | 9.71 | 51.50 | 73.65 | 25.67 | 0.8744 | 0.2265 |
| GOF [63] | 15.65 | 8.50 | 12.07 | 42.57 | 61.47 | 24.76 | 0.8646 | 0.2189 |
| PGSR [4] | 7.52 | 13.50 | 10.51 | 59.11 | 73.27 | 25.73 | 0.8761 | 0.2250 |
| QGS [69] | 8.53 | 10.02 | 9.28 | 55.15 | 73.66 | 24.37 | 0.8624 | **0.1994** |
| DN-Splatter [46] | 6.25 | 5.29 | 5.77 | 61.86 | 77.13 | 24.80 | 0.8549 | 0.2595 |
| **Ours** | **3.95** | **5.02** | **4.49** | **77.14** | **83.35** | **26.42** | **0.8874** | 0.2141 |

Table 2: Quantitative results of surface reconstruction on **ScanNet++** and **Replica** datasets. Our method outperforms other methods on most metrics. The best results are marked in **bold**.

|  | ScanNet++ | | | | | Replica | | | | |
|---|---|---|---|---|---|---|---|---|---|---|
|  | Acc↓ | Comp↓ | CD↓ | F1↑ | NC↑ | Acc↓ | Comp↓ | CD↓ | F1↑ | NC↑ |
| DUSt3R [50] | 9.70 | 6.64 | 8.17 | 38.17 | 75.27 | 7.52 | 7.18 | 7.35 | 44.89 | 68.19 |
| 3DGS [18] | 11.02 | 10.12 | 10.57 | 31.78 | 77.65 | 12.40 | 12.75 | 12.57 | 41.64 | 69.83 |
| 2DGS [15] | 6.89 | 6.52 | 6.7 | 53.46 | 88.20 | 9.02 | 12.05 | 10.54 | 52.91 | 82.28 |
| GOF [63] | 7.76 | 6.42 | 7.09 | 57.99 | 74.75 | 9.77 | 8.31 | 9.04 | 54.08 | 74.34 |
| PGSR [4] | 7.32 | 7.13 | 7.22 | 53.73 | 87.30 | 7.32 | 9.79 | 8.56 | 62.98 | 83.30 |
| DN-Splatter [46] | 4.88 | **3.44** | 4.16 | 75.86 | 82.06 | 4.95 | 6.25 | 5.60 | 68.12 | 80.80 |
| **Ours** | **3.86** | 3.46 | **3.66** | **82.78** | **90.52** | **2.80** | **5.45** | **4.13** | **81.90** | **89.88** |

**Qualitative Results:** Fig. 3 shows mesh comparisons. While 3DGS [18] produces rough surfaces, 2DGS/PGSR's [15, 4] flattened Gaussians create local smoothness but suffer from bending artifacts due to the missing of geometric constraints. Despite using depth and normal priors, DN-Splatter [46] still has inaccurate wall positioning in some scenes, and the lack of normal consistency causes roughness. Our method uniquely obtains both geometrically accurate planes and smooth surfaces with our planar and geometric priors, serving as a high-fidelity indoor reconstruction method. The quality of reconstruction is also demonstrated by the novel view rendering results shown in Fig. 4.

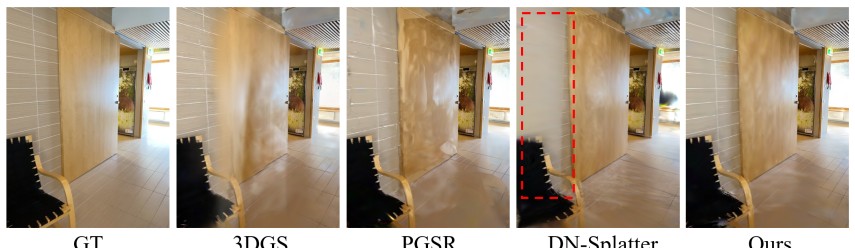

| GT | 3DGS | PGSR | DN-Splatter | Ours |

Figure 4: **Novel view synthesis comparison.** With the introduction of planar and geometric supervision, PlanarGS can effectively eliminate artifacts present in other methods.

## 4.2 Ablation Study

To validate the effectiveness of PlanarGS's components and the robustness of the pre-processing stage, we conducted ablation studies on the MuSHRoom and Replica datasets.

**Module Effectiveness:** We begin with an ablation on planar prior generation, comparing against the task-specific SOTA model ZeroPlane [29] and the vision-language foundation model Grounded-SAM [39] without cross-view fusion and geometric inspection. As in Tab. 3, in cases of inaccurate planar priors, co-planarity constraints may act on erroneous regions, leading to a reduction in reconstruction precision. A visual comparison of planar priors can be found in Fig. 5, where our planar priors effectively separate different planes and exclude non-planar clutter.

We further analyze the effect of each constraint in Gaussian optimization. When without the co-planarity constraint, as in Fig. 6, depth and normal priors provide geometric constraints only at the local scale, leading to surface irregularities in large plane regions. As can be seen from the Tab. 3, in the absence of geometric priors from pretrained models, the co-planarity constraint contributes more significantly to the reconstruction results. However, without geometric prior constraint, scale-aware supervision is missing, causing large planes under the co-planarity constraint to tilt and deviate as a whole, as in Fig. 6. Finally, when the consistency constraint between rendered normals and surface normals is absent, the flattened Gaussians fail to align with surfaces. This absence affects the normal accuracy of the mesh, displaying as rough surfaces particularly in regions without our co-planarity constraint, as shown in Tab. 3, Fig. 6.

Table 3: Ablation of module effectiveness on the coffee room of **MuSHRoom** dataset. The last row denotes our full model with all components enabled.

| Pre-proc. Planar Prior | Opt. Constraints | | | Metrics | | | | |
|---|---|---|---|---|---|---|---|---|
| | Co-planar. | Geom. Prior | DN Consist. | Acc↓ | Comp↓ | CD↓ | F1↑ | NC↑ |
| ZeroPlane [29] | ✓ | ✓ | ✓ | 4.76 | 5.73 | 5.25 | 62.49 | 82.41 |
| GroundedSAM [39] | ✓ | ✓ | ✓ | 2.95 | 3.92 | 3.43 | 80.22 | 84.66 |
| LP3 pipeline | - | ✓ | ✓ | 2.83 | 3.53 | 3.18 | 85.62 | 84.63 |
| LP3 pipeline | ✓ | - | ✓ | 4.63 | 5.01 | 4.82 | 73.96 | 80.62 |
| LP3 pipeline | - | - | ✓ | 6.93 | 6.26 | 6.59 | 64.11 | 74.42 |
| LP3 pipeline | ✓ | ✓ | - | 2.57 | 3.17 | 2.87 | 88.19 | 81.95 |
| LP3 pipeline | ✓ | ✓ | ✓ | 2.55 | 3.40 | 2.98 | 87.32 | 85.28 |

**Model Robustness:** To validate the robustness of our model, we perform extra ablations, as in Tab. 4. The multi-view foundation model can be substituted with VGGT [47], allowing this stage to be completed within a few seconds. We also replace the vision-language foundation model with YoloWorld [10] and SAM [19], and add "blackboard, sofa" in addition to regular prompts

for the Replica dataset. Both changes have minimal impact on the reconstruction results. These results highlight the robustness of our pipeline for Language-Prompted Planar Priors (LP3) and prior supervision, while indicating that improvements in foundation models may further boost the performance of our approach.

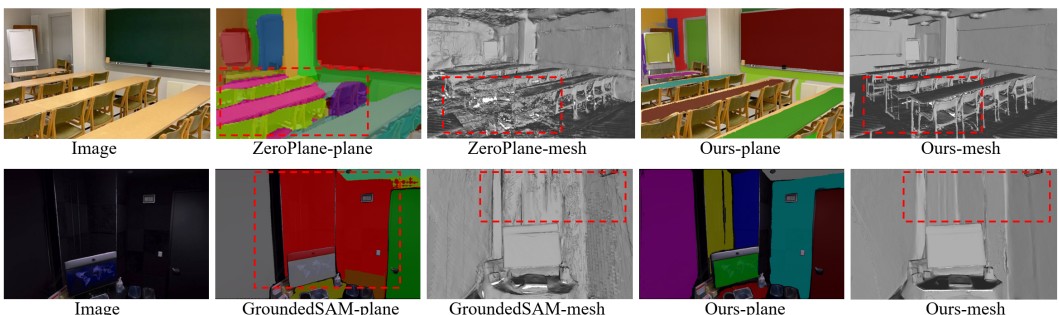

Figure 5: **Ablation of planar priors.** ZeroPlane tends to wrongly segment cluttered objects into planar regions, while using GroundedSAM without the pipeline for Language-Prompted Planar Priors (LP3) cannot distinguish different planes within a single object. Neither of them can provide reliable planar priors for Gaussian optimizing.

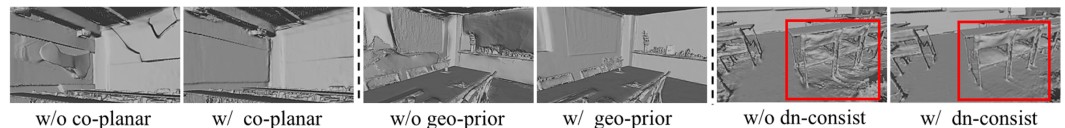

Figure 6: **Visualization of constraint effectiveness.** Left to right: results without and with co-planarity, geometric prior, and depth normal consistency constraints. Improvements are highlighted in red boxes. The enhancements show our proposed constraints lead to high-fidelity reconstruction.

Table 4: Ablation of model robustness on the **Replica** dataset. The last row shows the settings used for our model in the experiments.

| Pre-proccessing | | | Metrics | | | | |
|---|---|---|---|---|---|---|---|
| MV-FM | VL-FM | Prompts | Acc↓ | Comp↓ | CD↓ | F1↑ | NC↑ |
| VGGT [47] | GroundedSAM | regular | 3.96 | 6.32 | 5.14 | 75.41 | 88.85 |
| DUSt3R | YoloWorld [10]+SAM | regular | 2.92 | 5.56 | 4.24 | 80.96 | 89.04 |
| DUSt3R | GroundedSAM | more prompts | 2.88 | 5.50 | 4.19 | 81.43 | 89.32 |
| DUSt3R | GroundedSAM | regular | 2.80 | 5.45 | 4.13 | 81.90 | 89.88 |

# 5   Conclusion

We present a novel method, PlanarGS, that incorporates planar and multi-view depth priors into 3DGS, addressing the inaccurate reconstruction of large and texture-less planes commonly seen in indoor scenes. Extensive experiments on Replica, ScanNet++, and MuSHRoom datasets demonstrate that our method outperforms existing approaches in reconstruction. The language-prompted planar priors make our method flexible to multiple scenes in various domains like VR and robotics.

**Limitations:**   Our planar priors are only effective for detected plane regions, which work well in indoor environments dominated by large flat surfaces. However, these priors offer no improvement for the reconstruction of curved walls or natural outdoor scenes lacking man-made planar structures.

## Acknowledgments and Disclosure of Funding

This work was supported by the National Key R&D Program of China (2022YFB3903801) and the National Science Foundation of China (62073214).

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

# A  Technical Appendices and Supplementary Material

## A.1  Additional Inplementation Details

**Datasets:**  We conduct experiments with multi-view images from three indoor datasets. For the Replica dataset [44], we use eight scenes: office0-office4 and room0-room2, 100 views sampled from each scene. For ScanNet++ [60], we select sequences 8b5caf3398, b20a261fdf, 66c98f4a9b and 88cf747085 captured by a DSLR. Our experimental data on the MuSHRoom [40] consists of five short image sequences (coffee_room, classroom, honka, kokko, and vr_room) captured by an iPhone.

**Experiment Configuration:**  We employ an NVIDIA A6000 GPU with 48GB of VRAM for DUSt3R [50], which ensures each input group to DUSt3R contains 40 images, producing depth maps with better multi-view consistency. For the training process of all scenes, the depth normal consistency constraint $L_{dn}$ is introduced from 7000 iterations, the co-planarity constraint $L_p$ from 14000 iterations, while the prior depth and normal constraint $L_{rd}, L_{rn}$ from 7000 and 20000 iterations respectively. This training strategy ensures Gaussians are in relatively accurate positions before introducing planar priors, thereby reducing artifact generation. For novel view reconstruction, we use every eighth image from the input data as the test set, as in 3DGS.

## A.2  Additional Ablation Study

We present additional visualization of the ablation study on the MuSHRoom dataset here. Fig. 7-Fig. 9 sequentially show the ablation results of the co-planarity, geometric priors, and the depth normal consistency constraints.

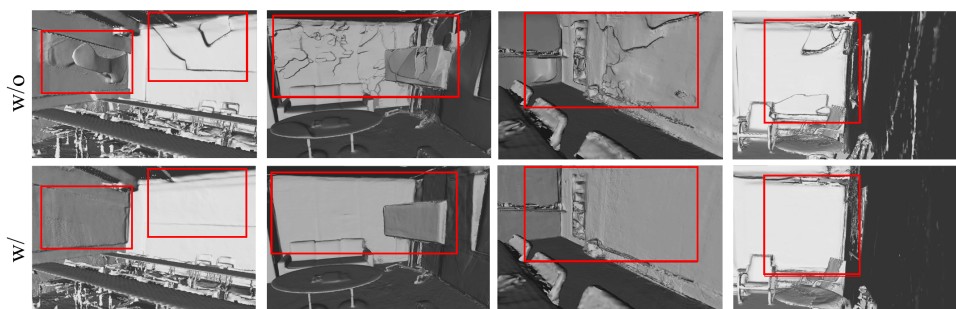

Figure 7: Ablation study of the **co-planarity constraint** on the **MuSHRoom** dataset. Improvements are highlighted in red boxes.

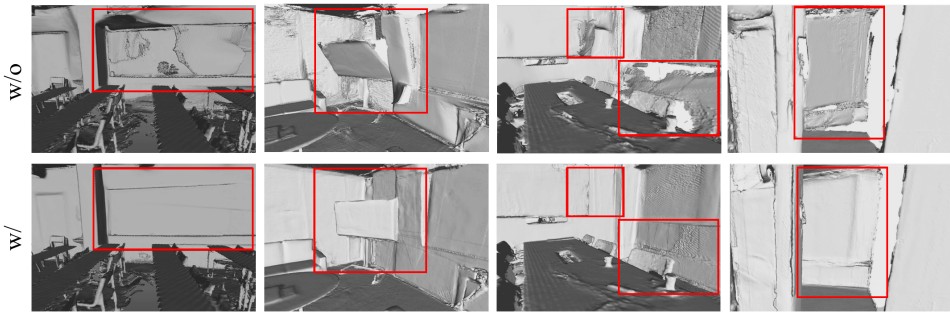

Figure 8: Ablation study of the **geometric prior constraint** on the **MuSHRoom** dataset. Improvements are highlighted in red boxes.

**Co-planarity Constraint:**  When the co-planarity constraint is absent, Gaussian optimization relies solely on view-wise depth and normal priors, only providing local geometric guidance of the current view. The lack of global guidance introduces geometric inconsistency into the reconstructed scene, causing large planes to fragment into multiple surfaces.

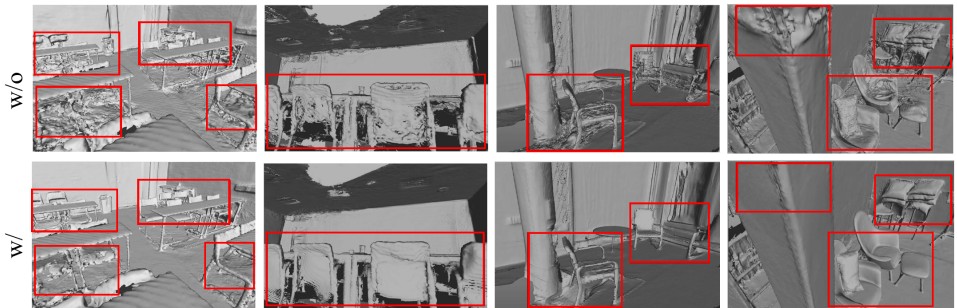

Figure 9: Ablation study of the **depth normal consistency constraint** on the **MuSHRoom** dataset. Improvements are highlighted in red boxes.

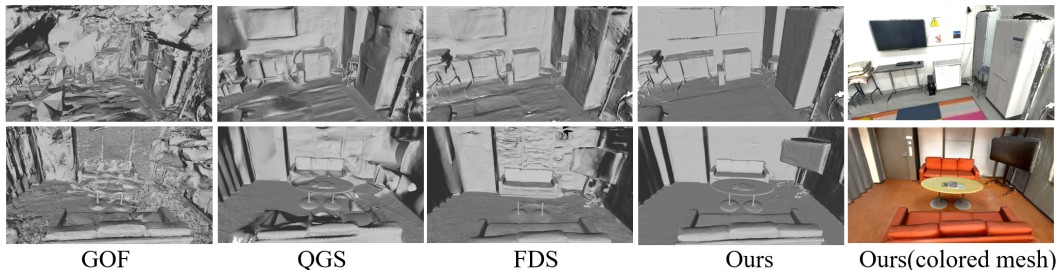

| GOF | QGS | FDS | Ours | Ours(colored mesh) |

Figure 10: Additional comparison of **mesh reconstruction** on the **MuSHRoom** dataset. Compared with these recent methods, our reconstruction results are noticeably smoother in planar regions.

**Geometric Prior Constraint:** Without depth priors from the multi-view foundation model [50], reconstruction in texture-less regions under photometric constraint lacks essential 3D structural and scale guidance, causing significant positional inaccuracy of Gaussians. In such cases, imposing co-planarity constraints on those artifacts may generate tilted or depth-inaccurate planes.

**Depth Normal Consistency:** The rotation of flattened Gaussians remains unconstrained without the depth normal consistency constraint, so they tend to be perpendicular to reconstructed surfaces. This results in surface roughness similar to that observed in meshes generated by 3DGS [18].

### A.3 Additional Results

**Surface Reconstruction:** We present a visual comparison of mesh reconstruction on the MuSH-Room dataset between our method and GOF [63], QGS [69], and FDS [7] in Fig. 10. Additionally, we present more extensive mesh reconstruction comparisons with the 3DGS [18], 2DGS [15], PGSR [4], and DN-Splatter [46] across the three datasets in Fig. 13-Fig. 15. PlanarGS achieves high-fidelity Gaussian reconstruction on all datasets, presenting significantly lower global and local errors and substantially outperforming previous methods.

**Novel View Synthesis:** We present the novel view synthesis results on the MuSHRoom dataset in Fig. 11, comparing our method with the ground truth, 3DGS [18], PGSR [4], and DN-Splatter [46]. It can be seen that in low-texture and reflective planar regions, other methods lack geometric constraints and produce artifacts in novel views due to the absence of planar priors, whereas our results are closer to the ground truth.

**Plane Masks and Rendering Results:** More comparisons with ZeroPlane [29] and Grounded-SAM [39] in terms of planar-prior generation are shown in Fig. 12. We also show visualizations of planar prior masks and rendering results of PlanarGS on three datasets in Fig. 16-Fig. 18. The first row displays the RGB images rendered by PlanarGS following Eq. 3, the second row shows the plane masks obtained by our pipeline for Language-Prompted Planar Priors (see Sec. 3.2), while the third and fourth rows exhibit the normal and depth maps rendered by PlanarGS through Eq. 7 and Eq. 9, respectively. Our method achieves robust planar prior masks and accurate RGB and geometric rendering, maintaining geometric consistency without compromising the rendering and reconstruction accuracy of Gaussian splatting.

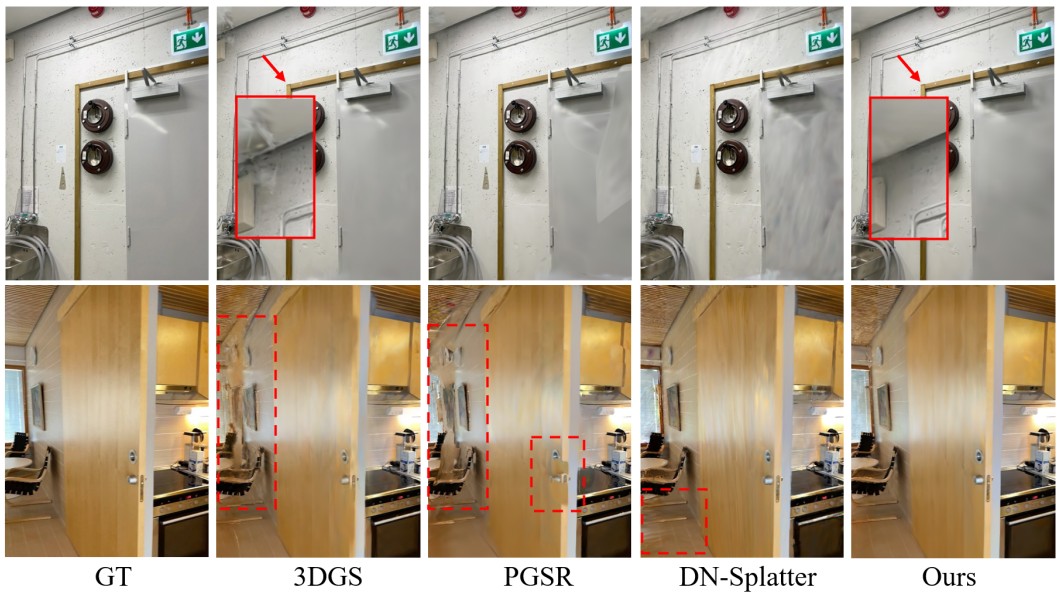

| GT | 3DGS | PGSR | DN-Splatter | Ours |

Figure 11: **Additional novel view synthesis visualization.** The highlighted regions are **enlarged** or **marked** with red boxes.

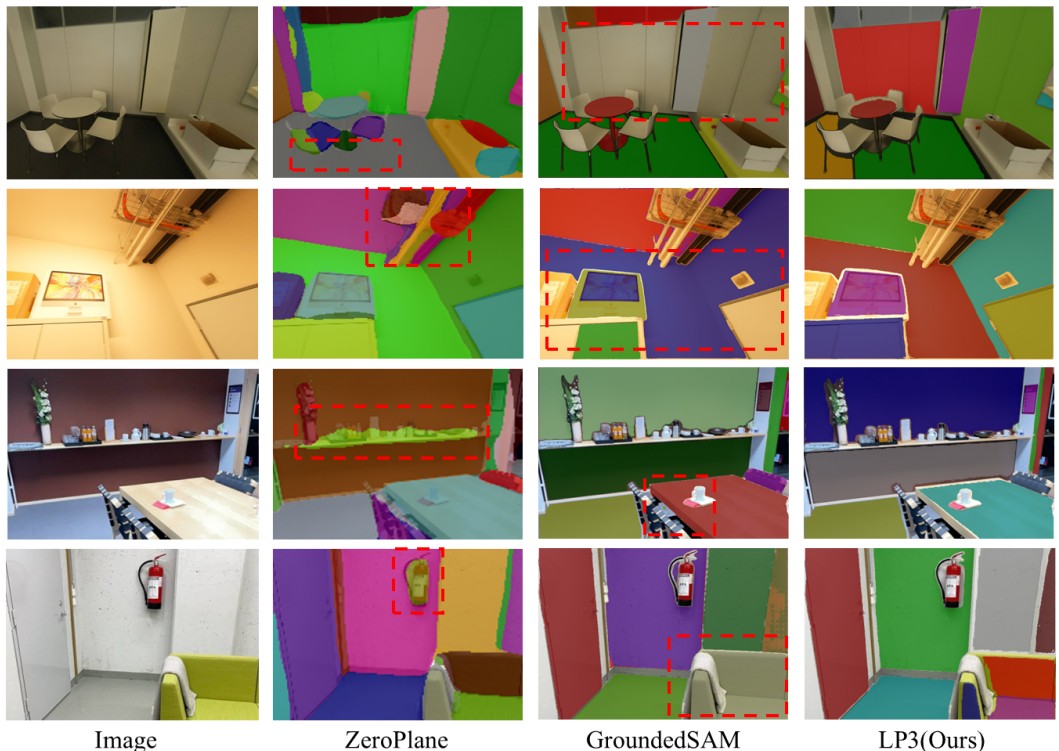

| Image | ZeroPlane | GroundedSAM | LP3(Ours) |

Figure 12: **Comparison of planar priors.** ZeroPlane fails to distinguish fine details or curved surfaces as non-planar regions, while GroundedSAM suffers from plane omission and merges planar regions.

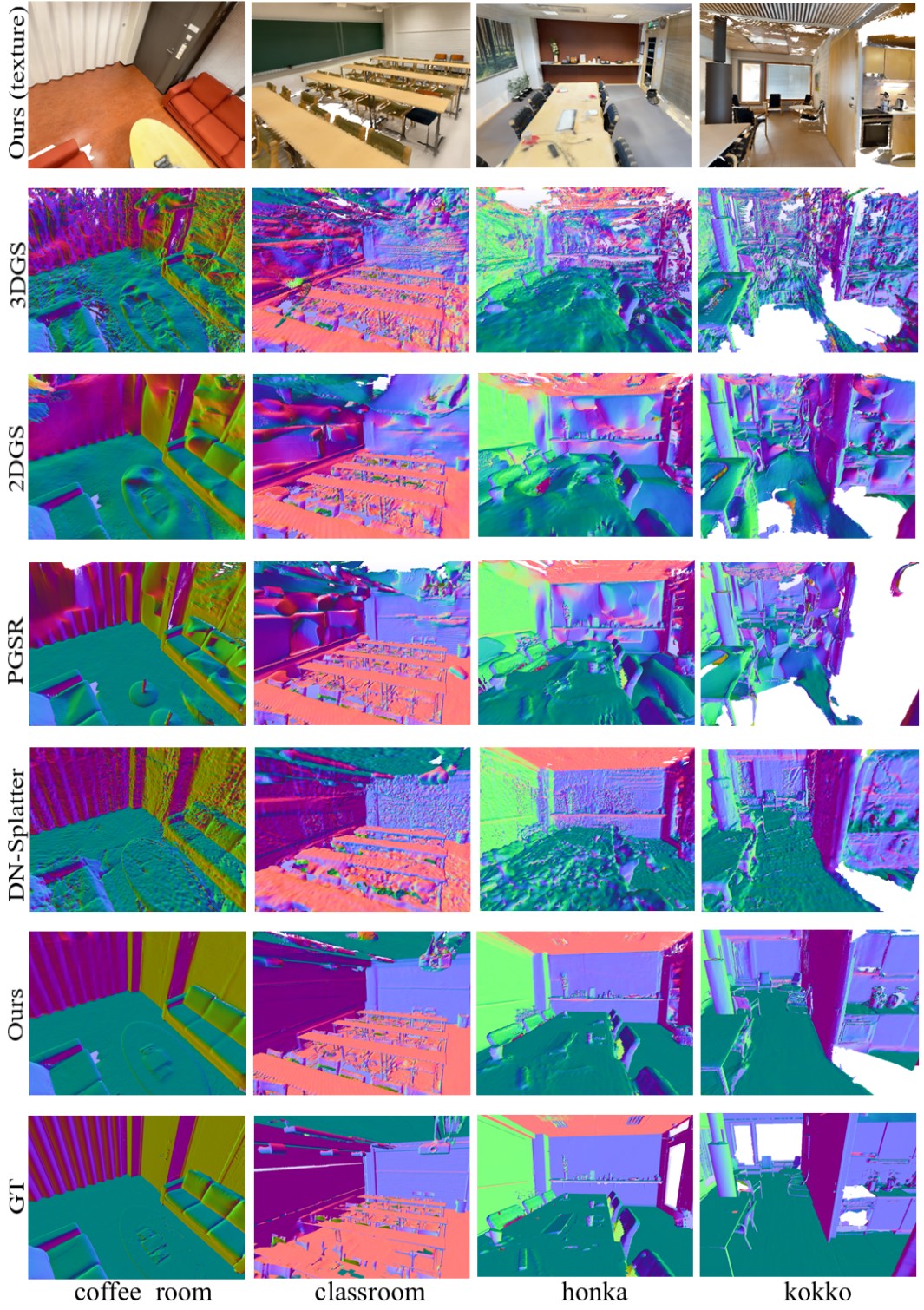

Figure 13: Additional qualitative comparison on the **MuSHRoom** dataset. We color the mesh according to surface normal directions to facilitate visual assessment of planarity consistency. Our meshes colored with textures are presented in the first row as references.

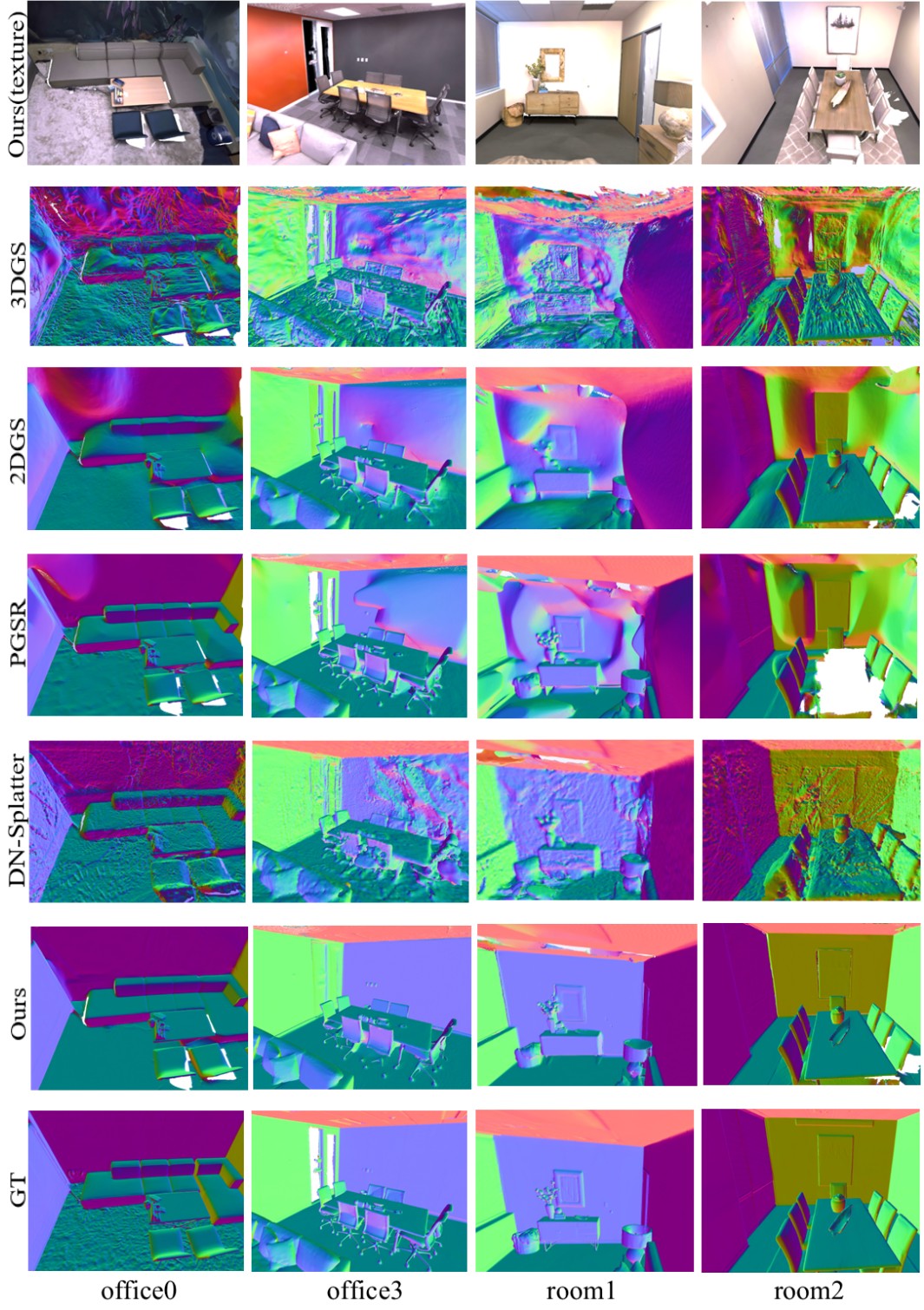

Figure 14: Additional qualitative comparison on the **Replica** dataset. We color the mesh according to surface normal directions to facilitate visual assessment of planarity consistency. Our meshes colored with textures are presented in the first row as references.

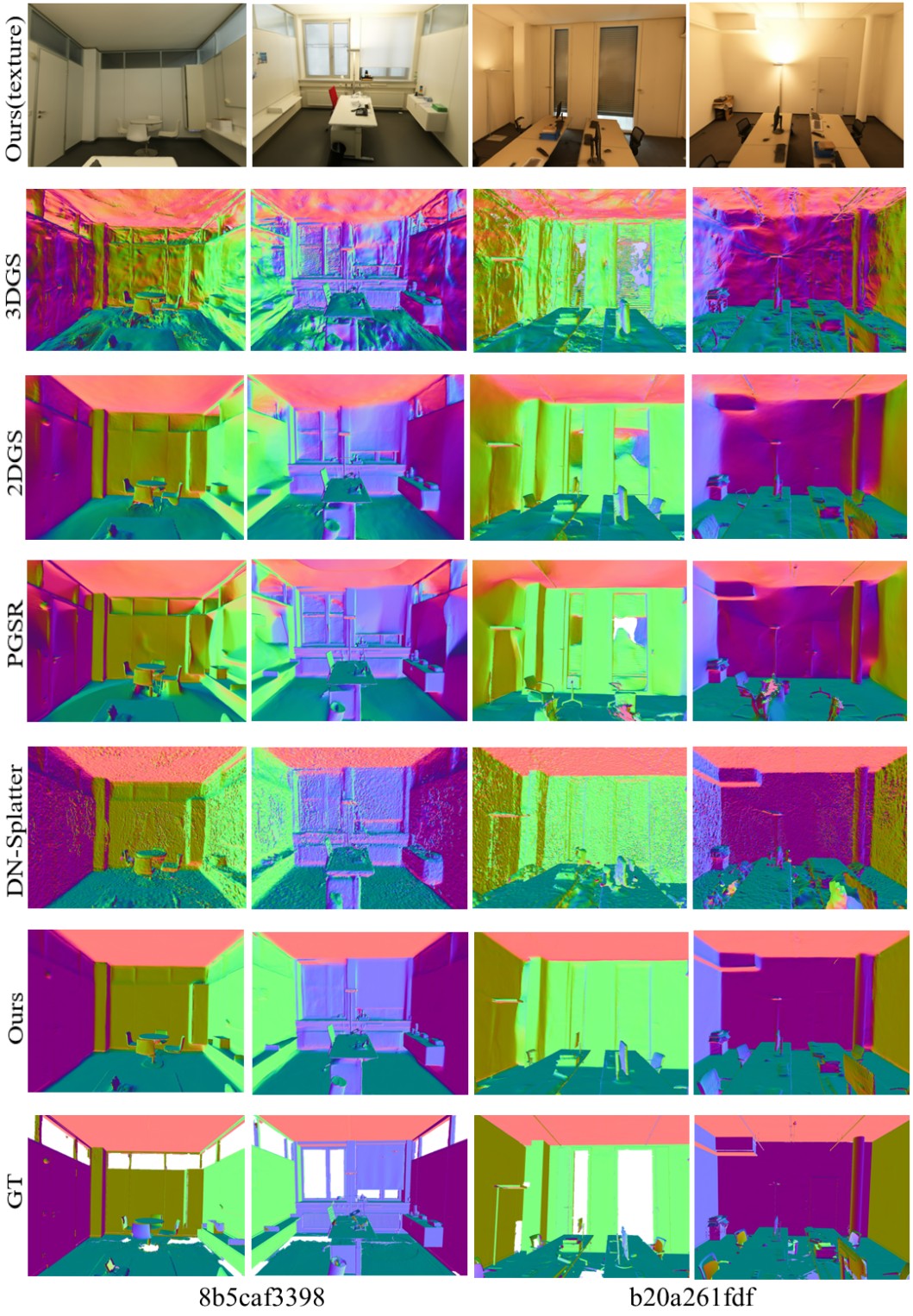

Figure 15: Additional qualitative comparison on the **ScanNet++** dataset. We color the mesh according to surface normal directions to facilitate visual assessment of planarity consistency. Our meshes colored with textures are presented in the first row as references.

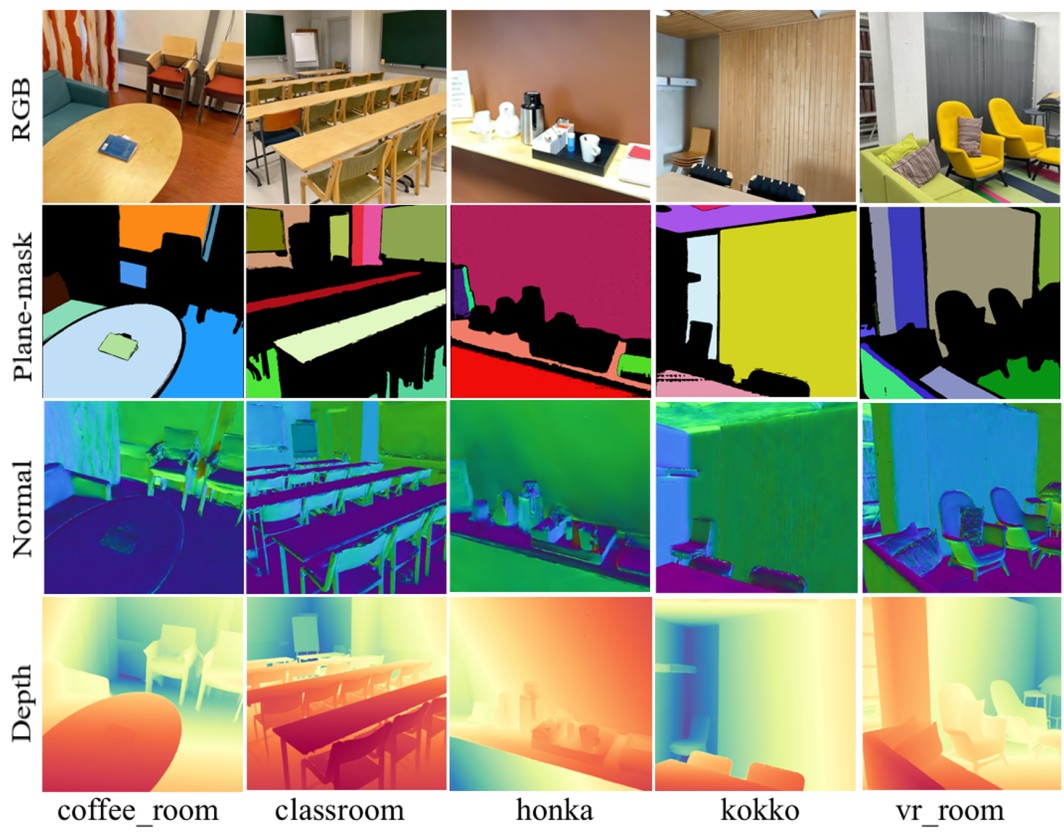

Figure 16: Rendering and plane-mask results on the **MuSHRoom** dataset.

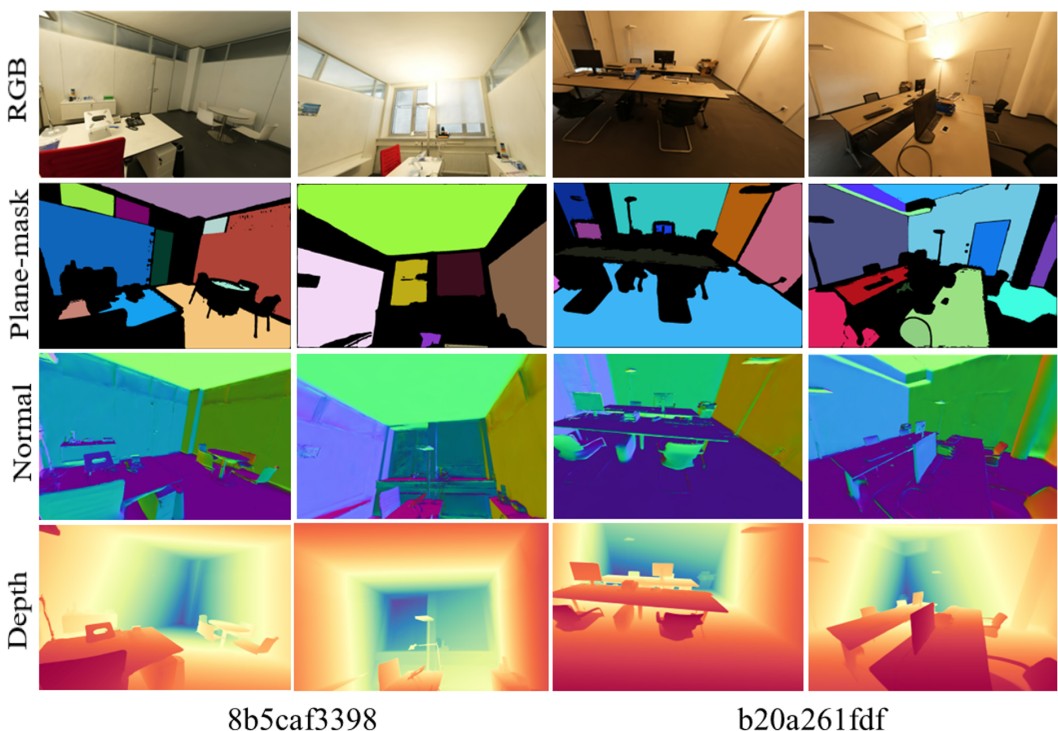

Figure 17: Rendering and plane-mask results on the **ScanNet++** dataset.

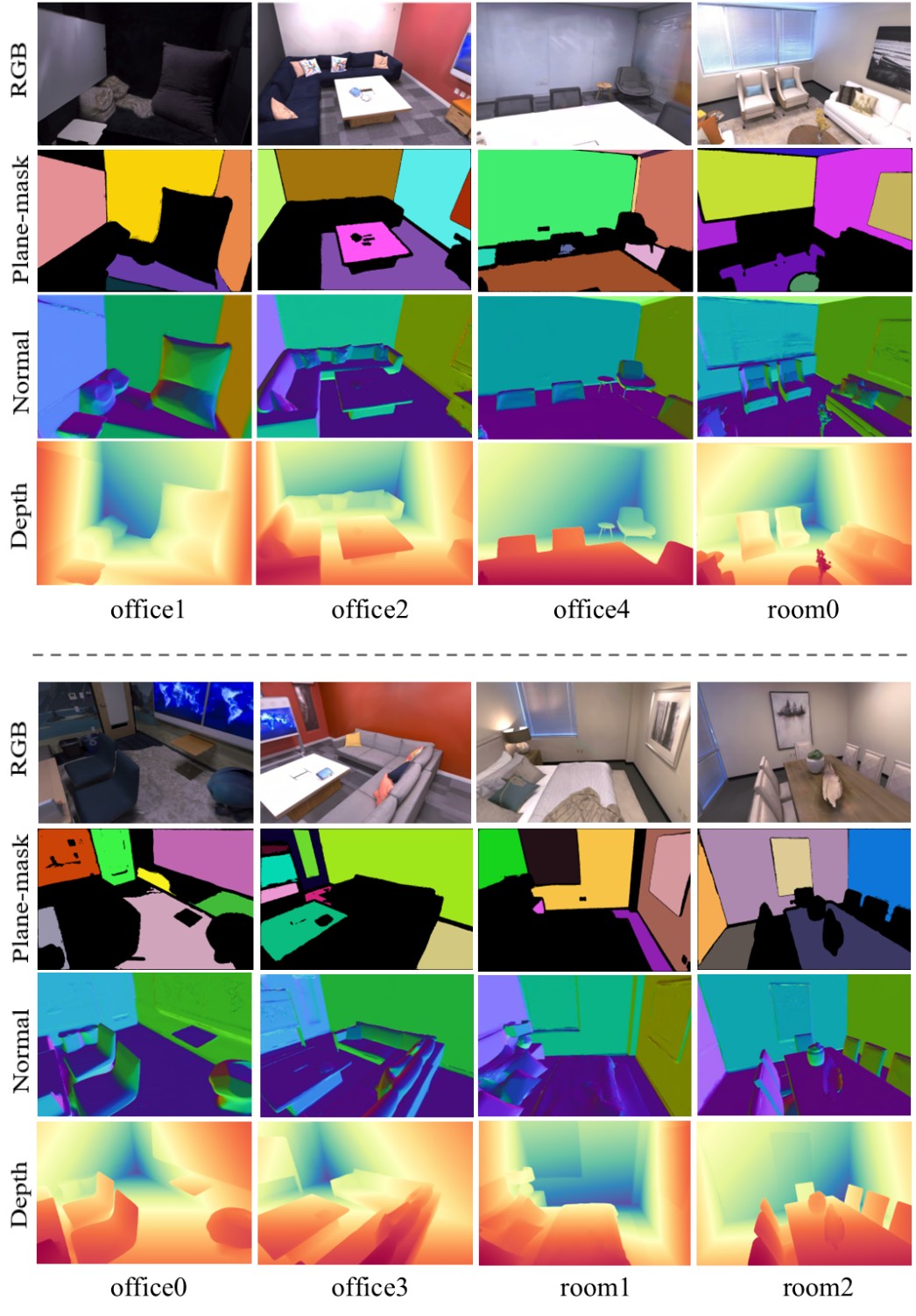

Figure 18: Rendering and plane-mask results on the **Replica** dataset.

