# OpenReview forum: "PlanarGS: High-Fidelity Indoor 3D Gaussian Splatting Guided by Vision-Language Planar Priors"
_NeurIPS.cc/2025/Conference — NeurIPS 2025 poster_

### Official Review · Reviewer_ew1g · 2025-06-30

**Clarity:** 3
**Significance:** 3
**Originality:** 2
**Rating:** 4
**Confidence:** 4

**Summary:**

The paper introduces PlanarGS, an new 3D Gaussian Splatting (3DGS) framework specifically designed to enhance 3D reconstruction in indoor environments, which often feature large, low-texture areas problematic for standard 3DGS. PlanarGS addresses this by incorporating semantic planar and geometric priors. It first employs a vision-language model to identify planar regions, refining these detections through cross-view fusion. Subsequently, the 3D Gaussians are optimized with two novel supervision terms: one to enforce planar consistency based on these detected regions, and another to guide the Gaussians using depth and surface-normal cues extracted from a separate 3D network. This holistic approach improves geometric accuracy and detail, leading to superior 3D surface reconstruction compared to existing methods.

**Questions:**

1. What are the main reason for being better than DNSplatter if not the coplanarity?
2. How does the proposed plane segmentation method compare with other ones?
3. What is the performance of GOF on these datasets?
4. What is the photometric accuracy on the evaluated datasets?

I am open to improving my rating, however, all these points are important to me to do so.

**Ethical Concerns:**

["NO or VERY MINOR ethics concerns only"]

**Final Justification:**

The authors addressed my concerns in their rebuttal.

**Limitations:**

Yes

**Paper Formatting Concerns:**

No concerns.

**Quality:**

3

**Strengths And Weaknesses:**

--- Strengths ---

The proposed method seems to improve geometric accuracy compared to the recent DN-Splatter.
However, the reason for this is unclear and there are questionable parts in the paper that should be clarified.

--- Weaknesses ---

- Table 4 suggests that the co-planarity constraints offer only minimal, albeit consistent, improvement. This leads to a question about the primary reasons for the reported performance gains, especially since DN-Splatter employs similar depth and normal losses. This must be explained.
- The paper proposes a multi-plane segmentation pipeline that's never evaluated in isolation or compared against baselines. Given the many existing multi-plane estimation methods (e.g., PlaneSegNet, PlaneRCNN, PlaneSeg, SPL-PlaneTR, PlaneSAM), the authors need to justify why their specific method was chosen for supervising 3DGS, particularly concerning computational cost (needs running VLLMs and Grounded DINO on many images in contrast to, e.g., running a simple plane region growing on a depth map). I do not say that the proposed method may not be more accurate than baselines. However, this must be verified to make the plane segmentation pipeline a contribution.
- Missing baselines from the tables: [a,b]. GOF [b] is the current state-of-the-art in surface reconstruction from 3DGS, so this must be added to the comparisons. [a] seems to be solving a similar problem as the proposed method.
- The tables only report the geometric accuracy and photometric ones. It would be important to verify that the photometric accuracy does not decrease.
- The paper is using different datasets that most of the 3DGS geometric reconstruction papers do that makes it hard to compare with results in other papers. It would help if the authors provide results on Tanks and Temples or DTU too.

---

> ### Author Rebuttal · Authors · 2025-07-31
>
> - **Q1:** What are the main reasons for being better than DNSplatter if not the coplanarity?
>
> - **A1:**  We would like to **clarify** that the performance gain brought by our co-planarity constraint is **not minimal** and **could not directly compare** to any specific component in DN-Splatter. First, as shown in the ablation study (**Table 4** in the main paper), removing the coplanar constraint leads to a **7%** increase in Chamfer Distance. Additionally, **Figure 5** in the Supplementary Material visually demonstrates the improvement in holistic planar smoothness brought by this constraint.
>
>   We have **included additional ablation studies**, which show that even **without geometric priors**, the coplanar constraint **significantly improves** reconstruction quality. This demonstrates the effectiveness of the planar prior.
>
>   |               | Acc $\downarrow$ | Comp $\downarrow$ | Prec $\uparrow$ | Recall $\uparrow$ | CD $\downarrow$ | F-score $\uparrow$ | Normal C. $\uparrow$ |
>   | :-----------: | :--------------: | :---------------: | :-------------: | :---------------: | :-------------: | :----------------: | :------------------: |
>   | w/o co-planr  |       2.83       |       3.53        |      88.92      |       82.55       |      3.18       |       85.62        |        84.63         |
>   | w/o geo-prior |       4.63       |       5.01        |      75.7       |       72.3        |      4.82       |       73.96        |        80.62         |
>   |   w/o both    |       6.93       |       6.26        |      64.11      |       65.22       |      6.59       |       64.11        |        74.42         |
>   |  full model   |       2.55       |        3.4        |      91.59      |       83.43       |      2.98       |       87.32        |        85.28         |
>
>   Additionally, our method is built upon the **PGSR codebase**, while DN-Splatter follows a substantially different pipeline. As stated in lines 278–280 of the main paper, DN-Splatter lacks normal consistency constraints. Moreover, its normal priors are derived from a monocular normal estimation model, rather than a multi-view foundation method. Its mesh extraction process also differs from ours. These differences **collectively** lead to DN-Splatter performing worse than even our method without the co-planarity constraint.
> - **Q2**: How does the proposed plane segmentation method compare with other ones?
>
> - **A2:** We have added **quantitative results** for multi-view planar segmentation on the EVOPS Benchmark, for the living-room and office scenes from the ICL dataset. The table includes comparison with the baseline method **PlaneRecNet**.
>
> |             | precision $\uparrow$ | recall $\uparrow$ | f-score $\uparrow$ |
> |:-----------:|-------------------|----------------|-----------------|
> | PlaneRecNet |             0.534 |          0.364 |           0.416 |
> | Ours        |             **0.582** |          **0.527** |           **0.538** |
>
> Many of the multi-plane estimation methods you mentioned, such as PlaneSeg and PlaneSAM, are **not open-sourced** or without checkpoints, making it infeasible to directly compare with them. As shown in the table, our zero-shot open-vocabulary plane segmentation model **outperforms** the strong baseline model **trained explicitly for plane segmentation**. We also obtained excellent qualitative results, which we regretfully cannot include here due to rebuttal constraints.
> We would like to re-explain our reasons for using the foundation segmentation model, as stated in lines 42–48 of the Introduction. The large-scale training and open-vocabulary input of advanced foundation models make them more **flexible** than task-specific models, resulting in better segmentation **quality** and stronger **cross-scene** generalization. Moreover, the depth priors utilized in our pipeline are **low-resolution results** from DUSt3R. To avoid compromising the accuracy of pixel-level segmentation, we do not use them directly for segmentation, but rather as a source of cross-view geometric information.
> - **Q3:** What is the performance of GOF on these datasets?
>
> - **A3:** We provide reconstruction results including **GOF** on the three datasets, as shown below.
>
> replica
>
> | Method       | L1↓   | Rel↓  | PSNR↑ | SSIM↑ | RMSE↓ | LPIPS↓ | Acc↑   |
> |--------------|-------|-------|-------|-------|-------|--------|--------|
> | 3DGS         | 12.40 | 12.75 | 43.17 | 40.24 | 12.57 | 41.64  | 69.83  |
> | 2DGS         | 9.02  | 12.05 | 58.30 | 48.45 | 10.54 | 52.91  | 82.28  |
> | PGSR         | 7.32  | 9.79  | 68.64 | 58.22 | 8.56  | 62.98  | 83.30  |
> | DN-Splatter  | 4.95  | 6.25  | 68.76 | 67.56 | 5.60  | 68.12  | 80.80  |
> | **Ours**     | **2.80** | **5.45** | **89.28** | **75.67** | **4.13** | **81.90** | **89.88** |
> | GOF          | 9.77  | 8.31  | 51.26 | 57.62 | 9.04  | 54.08  | 74.34  |
> | Dust3r_mesh  | 7.52  | 7.18  | 41.06 | 49.84 | 7.35  | 44.89  | 68.19  |
>
> scannet++
>
> | Method       | L1↓   | Rel↓  | PSNR↑ | SSIM↑ | RMSE↓ | LPIPS↓ | Acc↑   |
> |--------------|-------|-------|-------|-------|-------|--------|--------|
> | 3DGS         | 9.98  | 10.65 | 38.27 | 42.71 | 10.32 | 40.68  | 69.74  |
> | 2DGS         | 6.65  | 8.48  | 57.90 | 56.55 | 7.56  | 57.17  | 82.65  |
> | PGSR         | 6.77  | 7.63  | 58.42 | 59.29 | 7.20  | 58.79  | 83.07  |
> | DN-Splatter  | 3.80  | 2.46  | 79.84 | 91.46 | 3.13  | 85.22  | 82.30  |
> | **Ours**     | **3.41** | **2.42** | **85.02** | **91.18** | **2.91** | **87.96** | **89.37** |
> | GOF          | 7.76  | 6.42  | 52.70 | 64.52 | 7.09  | 57.99  | 74.75  |
> | Dust3r_mesh  | 7.96  | 5.24  | 41.94 | 64.46 | 6.60  | 50.69  | 75.51  |
>
> mushroom
>
> | Method       | L1↓   | Rel↓  | PSNR↑ | SSIM↑ | RMSE↓ | LPIPS↓ | Acc↑   |
> |--------------|-------|-------|-------|-------|-------|--------|--------|
> | 3DGS         | 12.01 | 11.85 | 35.84 | 41.76 | 11.92 | 38.53  | 62.00  |
> | 2DGS         | 9.16  | 10.27 | 52.31 | 50.84 | 9.71  | 51.50  | 73.65  |
> | PGSR         | 7.52  | 13.50 | 62.68 | 56.33 | 10.51 | 59.11  | 73.27  |
> | DN-Splatter  | 6.25  | 5.29  | 57.69 | 66.72 | 5.77  | 61.86  | 77.13  |
> | **Ours**     | **3.95** | **5.02** | **79.89** | **74.65** | **4.49** | **77.14** | **83.35** |
> | GOF          | 15.65 | 8.50  | 34.09 | 56.97 | 12.07 | 42.57  | 61.47  |
> | Dust3r_mesh  | 8.01  | 6.22  | 54.25 | 61.03 | 7.11  | 57.31  | 72.96  |
>
>   It can be clearly seen that our method consistently **outperforms** GOF by a large margin. In fact, GOF was proposed before PGSR and has already **been surpassed by PGSR** on multiple datasets. Moreover, the mesh extraction process of GOF requires **significant GPU resources**, which is another reason we did not include it in the main comparisons of our paper.
>
> - **Q4:** What is the photometric accuracy on the evaluated datasets?
>
> - **A4:** We report photometric accuracy on the Mushroom and ScanNet++ datasets, and observe comparable performance with the baseline methods.
>
> Photometric Metrics on Four Scenes of the ScanNet++ Dataset
>
> |             | PSNR $\uparrow$ | SSIM $\uparrow$ | LPIPS $\downarrow$ |
>   | :---------: | :-------------: | :-------------: | :----------------: |
>   |    3DGS     |      24.93      |     0.8747      |       0.2419       |
>   |    2DGS     |      25.59      |     0.8804      |       0.2394       |
>   |     GOF     |      25.68      |     0.8808      |       0.2363       |
>   |    PGSR     |      25.43      |     0.8799      |       0.2379       |
>   | DN-Splatter |      22.99      |     0.8401      |       0.1984       |
>   |    Ours     |      24.64      |     0.8700      |       0.2536       |
>
> Photometric Metrics on Four Scenes of the Mushroom Dataset
>
> |             | PSNR $\uparrow$ | SSIM $\uparrow$ | LPIPS $\downarrow$ |
>   | :---------: | :-------------: | :-------------: | :----------------: |
>   |    3DGS     |      25.79      |     0.8775      |       0.2059       |
>   |    2DGS     |      25.67      |     0.8744      |       0.2265       |
>   |     GOF     |      24.76      |     0.8646      |       0.2189       |
>   |    PGSR     |      25.73      |     0.8761      |       0.2250       |
>   | DN-Splatter |      24.81      |     0.8537      |     **0.1394**     |
>   |    Ours     |    **26.42**    |   **0.8874**    |       0.2141       |
>
>   The challenging test set of Scannet++ consists of images with viewpoints that significantly deviate from the training sequences. For Mushroom, we follow the protocol used in 3DGS and sample one out of every eight images from the sequence to construct the test set.
>
> - **Q5:** It would help if the authors provide results on Tanks and Temples or DTU too.
>
> - **A5:** We have conducted experiments on the Meetingroom and Barn scenes from the **TnT** dataset as below. In both scenes, our method achieves **high-quality** reconstructions. ( "-" indicates that the method **failed** to generate a mesh.)
>
>   |             | 3DGS  | SuGaR | 2DGS  | GOF  | PGSR | DN-Splatter |   Ours   |
>   | :---------: | :---: | :---: | :---: | :--: | :--: | :---------: | :------: |
>   |    Barn     | 0.13  | 0.14  | 0.36  | 0.51 | 0.66 |      -      | **0.70** |
>   | Meetingroom | 0.01  | 0.15  | 0.16  | 0.28 | 0.29 |      -      | **0.55** |
>   |    time     | 14.3m |  2h   | 34.2m |  2h  | 1.2h |    1.1h     |    1h    |
>
>   We did not conduct experiments on the **object-level DTU dataset**, following the practice of other indoor reconstruction works. Although the TnT dataset is scene-level, it contains **only one indoor scene** with ground truth—the Meetingroom. Additionally, we evaluated our method on the **outdoor man-made structure** Barn to demonstrate the potential of our approach to generalize to outdoor reconstruction with plane regions.

---

> > ### Comment · Reviewer_ew1g · 2025-08-01
> >
> > I am happy with these results. Question: does the runtime include plane segmentation on Barn/Meetingroom? If not, how long does it take?

---

> > > ### Author Response · Authors · 2025-08-01
> > >
> > > Thank you for your response. The reported runtime in the table does not include the time for plane segmentation. On a TNT scene (around 370 images), our plane segmentation takes **approximately 130 seconds**, which is comparable to the original GroundedSAM.

---

### Official Review · Reviewer_i6Sf · 2025-07-02

**Clarity:** 3
**Significance:** 3
**Originality:** 3
**Rating:** 5
**Confidence:** 4

**Summary:**

This paper proposes PlanarGS, a method that incorporates plane priors detected by vision-language models with multi-view depth-normal geometric priors to enhance indoor 3D Gaussian Splatting. The approach improves the accuracy and completeness of plane detection through cross-view plane fusion and geometric verification. It introduces supervision strategies specifically for planar and non-planar regions, including Gaussian flattening, coplanarity constraints, and depth-normal consistency. Experiments conducted on multiple real and synthetic indoor datasets demonstrate that the method significantly outperforms existing SOTA methods.

**Questions:**

If the multi-view images of a scene are captured from discrete viewpoints rather than a continuous sequence, how should the neighboring views be selected in the cross-view fusion process?

**Ethical Concerns:**

["NO or VERY MINOR ethics concerns only"]

**Final Justification:**

The authors’ rebuttal has addressed my concerns. After carefully reading and considering the other reviews and the rebuttal, I do not see major issues and will maintain my original score (accept). However, I suggest the authors include experimental results on more scenes in the paper.

**Limitations:**

yes

**Quality:**

4

**Strengths And Weaknesses:**

## Strengths

1. The proposed method presents a well-designed and technically sound framework for integrating plane priors into 3D scene reconstruction, contributing substantially on a technical level.

2. Qualitative results show high-quality reconstruction of indoor scenes, with very smooth planar regions; the quantitative results also demonstrate significant improvements over previous methods, which is highly convincing.

3. The paper includes comprehensive ablation studies to validate the effectiveness of each design component.

## Weaknesses

1. The method involves a relatively complex pipeline with many steps and heuristic components—for example, using Grounded-SAM for initial plane segmentation, DUST3R for depth and normal extraction, and Canny edge detection for computing the mask of low-texture regions. Some steps also involve hyperparameter choices, such as the confidence threshold in GroundingDINO. Therefore, the robustness of the method remains to be discussed.

2. Minor issues: "SFM" is better to be written as "SfM"; "visual" in line 299 should be corrected to "vision" for consistency.

---

> ### Author Rebuttal · Authors · 2025-07-31
>
> We appreciate your positive feedback on our method and experiments. We address your comments in our below responses.
>
> - **Q1:** The method involves a relatively complex pipeline with many steps and heuristic components, together with hyperparameter choices. Therefore, the robustness of the method remains to be discussed.
> - **A1:** PlanarGS **is** a robust method. The threshold of confidence and canny mask, as well as all the other various hyperparameters in the 3D reconstruction pipeline, are **kept consistent** across all scenes and datasets.
> - **Q2:** Minor issues: "SFM" is better to be written as "SfM"; "visual" in line 299 should be corrected to "vision" for consistency.
> - **A2:** Thank you for your careful reading and correction. We will revise them in future versions of the paper.
> - **Q3:** If the multi-view images of a scene are captured from discrete viewpoints rather than a continuous sequence, how should the neighboring views be selected in the cross-view fusion process?
> - **A3:** For unordered input, we reorder them as a sequence before our plane segmentation, by calculating the field-of-view (FOV) similarity between each frames and solve the graph traversal problem.

---

> > ### Comment · Reviewer_i6Sf · 2025-08-06
> >
> > Thanks for the authors’ rebuttal, which has addressed my concerns. After carefully reading and considering the other reviews and the rebuttal, I do not see major issues and will maintain my original score (accept). However, I suggest the authors include experimental results on more scenes in the paper.

---

### Official Review · Reviewer_q67a · 2025-07-03

**Clarity:** 2
**Significance:** 2
**Originality:** 2
**Rating:** 4
**Confidence:** 4

**Summary:**

The paper proposes the PlanarGS, which leverages vision-language plane priors and geometric priors to guide 3D Gaussian splatting, addressing the issue of geometric ambiguity in 3D reconstruction of low-texture planar regions in indoor environments. Experimental results demonstrate that its performance outperforms existing methods.

**Questions:**

🤔

1. The experiments claim to effectively address indoor environments with numerous low-texture planes, yet the presented cases are overly simplistic. I am curious: would the performance of the proposed method be affected in environments with a large number of curved furniture pieces?
2. The work uses Grounded SAM for segmentation and relies on DUSt3R for geometric verification and refinement. However, in scenarios with insufficient information (e.g., specular reflections, backlit environments), could this approach introduce additional errors, leading to harm to model robustness?
3. The paper only compares 4 methods. To my knowledge, other methods such as Manhattan achieve comparably competitive visualization results. The absence of comparisons with such works would significantly undermine the persuasiveness of the proposed method.

**Ethical Concerns:**

["NO or VERY MINOR ethics concerns only"]

**Final Justification:**

The author seems to have addressed my concerns. However, since this submission was randomly assigned to me and it is not my area of expertise, despite my best efforts in reviewing, I still suggest that the AC take into account the opinions of other reviewers more. Thank you.

**Limitations:**

yes

**Quality:**

3

**Strengths And Weaknesses:**

**Strengths😊**
1. The paper primarily introduces planar constraints and geometric priors to enhance geometric accuracy in low-texture regions, resolving the performance limitations of existing methods (e.g., 2D/3DGS, PGSR) when handling large-scale textureless planes.
2.  Experiments cover multiple types of indoor datasets, with both quantitative and qualitative results significantly outperforming baseline methods such as 3DGS and PGSR, validating the effectiveness of the framework.

**Weaknesses😭**
1. While ablation studies verify the effectiveness of individual components, they fail to compare the impact of different vision-language models on plane detection, leading to insufficient verification of robustness.
2. Multi-view geometric priors (e.g., DUSt3R) incur high computational costs, which may hinder real-time performance. No mention is made of efficiency optimizations for practical deployment.
3. Plane priors rely on detection results from pre-trained models. How is the introduction of additional errors in this process prevented?

---

> ### Author Rebuttal · Authors · 2025-07-31
>
> - **Q1:** Lack of comparison of the impact of different vision-language models on plane detection, leading to insufficient verification of robustness.
> - **A1:** We supplement **our plane segmentation method with quantitative comparisons against other approaches**.
>
> |             | precision $\uparrow$ | recall $\uparrow$ | f-score $\uparrow$ |
> |:-----------:|-------------------|----------------|-----------------|
> | PlaneRecNet |             0.534 |          0.364 |           0.416 |
> | Ours        |             **0.582** |          **0.527** |           **0.538** |
>
> As stated in our abstract, our method primarily focuses on improving the geometric reconstruction of 3D Gaussians in low-texture plane regions of indoor scenes. Therefore, we emphasize the surface reconstruction quality and related contributions. Here, we supplement **our plane segmentation method with quantitative comparisons against other approaches**. Unfortunately, due to the rebuttal limitations, we are unable to present visual results. We will consider conducting the robustness experiment you mentioned, replacing GroundedSAM with other vision-language foundation models.
> - **Q2:** Multi-view geometric priors (e.g., DUSt3R) incur high computational costs, which may hinder real-time performance.
> - **A2:** Our work primarily addresses the issue of supervising 3DGS using pre-trained multi-view geometric models, whereas improving the foundation model itself is **beyond the scope of this paper**, as stated in the third contribution of our Introduction. Our method does not rely on a specific foundation model and can be robustly adapted to alternatives. The latest **VGGT** model enables reconstruction of **hundreds of views within seconds**. Here, we provide results on the Replica dataset after replacing DUSt3R with VGGT.
>     |                 | Acc $\downarrow$ | Comp $\downarrow$ | Prec $\uparrow$ | Recall $\uparrow$ | CD $\downarrow$ | F-score $\uparrow$ | Normal C. $\uparrow$ |
>   | :-------------: | :--------------: | :---------------: | :-------------: | :---------------: | :-------------: | :----------------: | :------------------: |
>   | DUSt3R-PlanarGS |       2.80       |       5.45        |      89.28      |       75.67       |      4.13       |       81.90        |        89.88         |
>   |  VGGT-PlanarGS  |       3.96       |       6.32        |      81.87      |       69.91       |      5.14       |       75.41        |        88.85         |
>
> Replacing the foundation model within the PlanarGS pipeline is feasible and results in only minor changes to the evaluation metrics. Future foundation models are expected to further enhance our reconstruction efficiency and performance.
> - **Q3:** Plane priors rely on detection results from pre-trained models. How is the introduction of additional errors in this process prevented?
> - **A3:** Our method **does not rely on highly accurate outputs** from the pre-trained models. Our strict **cross-view fusion** and **geometric inspection** modules significantly enhance the recall and precision of our open-vocabulary plane segmentation method, as described in Section 3.2 of the paper. Moreover, 3D reconstruction **inherently benefits from multi-view consistency**, and the **geometric priors** from the pre-trained model also provide strong supervision. As a result, occasional segmentation errors in individual views **barely** affect the overall reconstruction quality.
> - **Q4:** The presented cases are overly simplistic. Would the performance of the proposed method be affected in environments with a large number of curved furniture pieces?
>
> - **A4:** We have to clarify that the presented cases **are not** overly simplistic, and our method demonstrates **strong generalization ability** in cluttered scenes. Our dataset selection follows the **common practice in prior indoor 3DGS methods**, such as DN-Splatter. In particular, the Mushroom dataset covers a wide range of real-world complex indoor structures, as stated in Implementation Details. As you can see in Supplementary Material **Fig8,11**, **the scenes are cluttered with various objects and curved furniture**. By providing GroundedSAM with prompts related to planar objects, curved clutter and furniture are typically **not identified as plane regions**. Occasional segmentation or recognition errors can be effectively filtered out during **the Geometric Inspection** step by leveraging prior normal and depth information, as described in lines 181–186 of Section 3.2. These typically textured objects out of plane masks are then reconstructed with high fidelity through the **existing high-quality Gaussian surface reconstruction baseline**.
>
> - **Q5:** The work uses Grounded SAM for segmentation and relies on DUSt3R for geometric verification and refinement. However, in scenarios with insufficient information (e.g., specular reflections, backlit environments), could this approach introduce additional errors, leading to harm to model robustness?
>
> - **A5:** Our method is **particularly robust** in environments with insufficient information. The foundation models are trained on large-scale data and **demonstrate strong robustness** in regions with insufficient information, which is the key reason we incorporate them into our 3DGS pipeline. Unfortunately, we are unable to show the depth prior images from DUSt3R here. However, as shown in **Figure 3** of the main paper, our method performs well in challenging conditions: the first row features a completely dark scene, the third row contains strong wall reflections and overexposure, and the last row includes specular reflections from a screen. None of these factors compromises our reconstruction quality. Additionally, **Figures 11–13** in the supplementary material further demonstrate that our plane segmentation remains robust under varying lighting conditions. You can have a better look at our mesh results in our **video**.
>
> - **Q6:** The absence of comparisons with works such as Manhattan.
>
> - **A6:** I think you are referring to the **Manhattan-SDF** work here, and we don't think we have to compare with it. As we mentioned in the second paragraph of the Related Work Section, Manhattan-SDF is based on the time-consuming **NeRF** framework, and its MLP fusion approach cannot be directly transferred to Gaussian-based methods. Our method is an indoor scene improvement of the **3DGS** surface reconstruction approach, so we only compare it with Gaussian-based methods that have **comparable training times**.

---

> > ### Author Response · Authors · 2025-08-08
> > **The robustness of our plane detection pipeline**
> >
> > Regarding your concern about **the robustness of our plane detection pipeline**, we follow your suggestion and conduct an experiment on the Replica dataset using another advanced open-vocabulary object recognition model, Yolo-world[1], instead of GroundingDINO.
> >
> > |                    | Acc ↓ | Comp ↓ | Prec ↑ | Recall ↑ | CD ↓ | F-score ↑ | Normal C. ↑ |
> > | :----------------: | :---: | :----: | :----: | :------: | :--: | :-------: | :---------: |
> > |  Ours_Yolo-world   | 2.92  |  5.56  | 87.82  |  75.12   | 4.24 |   80.96   |    89.04    |
> > | Ours | 2.80  |  5.45  | 89.28  |  75.67   | 4.13 |   81.90   |    89.88    |
> >
> > Changing to another vision-language model, our method is still effective.
> >
> > [1] Cheng T, Song L, Ge Y, et al. Yolo-world: Real-time open-vocabulary object detection[C]//Proceedings of the IEEE/CVF conference on computer vision and pattern recognition. 2024: 16901-16911.

---

### Official Review · Reviewer_zbUe · 2025-07-23

**Clarity:** 3
**Significance:** 2
**Originality:** 1
**Rating:** 3
**Confidence:** 3

**Summary:**

This paper introduces planar and geometric depth priors into the 3D Gaussian Splatting (3DGS) framework to address challenges in indoor scene reconstruction, particularly in texture-less and large regions. The authors leverage pre-trained vision-language segmentation models for plane detection and utilize DUST3R to extract depth and surface normal information. They propose two supervision terms—a planar-prior loss and a geometric-prior loss—that guide the optimization of 3D Gaussians toward more accurate planar and depth-aware representations. Final mesh reconstructions are obtained using a TSDF fusion approach. Experiments on selected scenes from the Replica, ScanNet++, and MuSHRoom datasets demonstrate improved performance in reconstructing planar structures within indoor scenes.

**Questions:**

1. None of the comparisons include the original input images, making it difficult to qualitatively assess whether the reconstructions accurately reflect the real scenes. Visual comparison against the source images is essential for evaluating geometric and appearance fidelity.
2. The paper opts for a general-purpose vision-language segmentation model paired with DUST3R for surface normal estimation. However, this raises a key question: why not use a task-specific method designed for indoor surface normal estimation? Many recent methods can handle both structured and texture-less regions robustly without requiring text prompts. The use of vision-language models adds complexity and fragility without clear advantages in this domain.

3. It is unclear whether the segmentation prompts need to be modified for each new scene. If so, this drastically limits the scalability and practicality of the method. Scene-specific prompt engineering is not viable for real-world applications.

**Ethical Concerns:**

["NO or VERY MINOR ethics concerns only"]

**Final Justification:**

I appreciate the authors' detailed answers and all the additional experiments. After reading the response and other reviewers' comments, some of my concerns have been addressed. However, several issues remain. This paper works in a very narrow use-case, with all results shown in ideal scenarios with large planar areas, which are uncommon in indoor spaces due to decoration. The authors also admit in their rebuttal that texture reduces the benefits of their design. They haven't adequately addressed my concern about dependency on fundamental models - when these models fail, their method fails, making it less reliable than COLMAP. A handful of results isn't sufficient to draw conclusions about generalization and robustness. Additionally, while I appreciate the engineering efforts, I believe the paper lacks sufficient novelty. Therefore, I'll raise my score slightly, but still consider the paper below the bar for NeurIPS.

**Limitations:**

1. The method relies heavily on external models—particularly segmentation foundation models and DUST3R. Any failure or mismatch in these upstream models directly impacts the quality and stability of the final reconstruction.
2. The proposed pipeline is tailored for indoor environments with large, flat, texture-less surfaces. It is not easily extendable to more diverse indoor environments (e.g., with cluttered small-scale texture-less objects like books or paintings), let alone outdoor scenes, which also include planar regions but typically present very different geometric and appearance characteristics.

**Paper Formatting Concerns:**

no concerns.

**Quality:**

2

**Strengths And Weaknesses:**

Strengths:
1. The paper is clearly written and provides sufficient implementation details to enable reproducibility.
2. The idea of leveraging pre-trained foundation models to enhance 3D reconstruction is timely and relevant.
3. The authors introduce novel supervision strategies for both planar and non-planar regions during 3D Gaussian Splatting (3DGS) optimization, which improves the accuracy and robustness of the reconstruction results.

Weaknesses:
1. Limited Novelty: The paper primarily integrates existing pre-trained models into 3DGS optimization for a specific indoor reconstruction task. This integration feels more like an engineering effort than a contribution of fundamentally new techniques. The technical novelty is marginal.
2. The method targets indoor scenes with large, texture-less planar surfaces. However, this scope is quite narrow, and the method’s success depends on prior knowledge of such regions, which must be explicitly specified via prompts for segmentation. This reliance hinders generalization.
3. Heavy Dependence on Pretrained Models: The pipeline depends critically on the success of external models (e.g., segmentation and DUST3R). Failure in any of these steps can cause the entire system to break down, raising concerns about robustness. Compared to traditional methods like COLMAP, this approach may be less reliable.
4. The experimental benchmark is extremely limited—only 15 test scenes in total. For instance, just 2 scenes were selected from the 460-scene ScanNet++ dataset, both featuring white walls. This narrow and selective evaluation raises concerns about cherry-picking and makes it difficult to assess the general effectiveness of the method. Quantitative results are therefore not convincing at all.
5. Incomplete Comparisons: If the paper claims to advance 3D reconstruction, it should compare against strong reconstruction baselines such as COLMAP, DUST3R, and 3DGS-based methods like SUGAR. Furthermore, if the focus is on 3DGS, the paper should report rendering metrics and visual results, as 3DGS is specifically designed for high-quality rendering. Their omission is a major oversight.

---

> ### Author Rebuttal · Authors · 2025-07-31
>
> - **Q1:** Lack of qualitative and quantitative comparison for novel view synthesis
>
> - **A1:** We additionally report **photometric metrics** of both the ScanNet++ and Mushroom datasets.
>
>       Photometric Metrics on Four Scenes of the ScanNet++ Dataset
>
>   |             | PSNR $\uparrow$ | SSIM $\uparrow$ | LPIPS $\downarrow$ |
>   | :---------: | :-------------: | :-------------: | :----------------: |
>   |    3DGS     |      24.93      |     0.8747      |       0.2419       |
>   |    2DGS     |      25.59      |     0.8804      |       0.2394       |
>   |     GOF     |      25.68      |     0.8808      |       0.2363       |
>   |    PGSR     |      25.43      |     0.8799      |       0.2379       |
>   | DN-Splatter |      22.99      |     0.8401      |       0.1984       |
>   |    Ours     |      24.64      |     0.8700      |       0.2536       |
>
>       Photometric Metrics on Four Scenes of the Mushroom Dataset
>
>   |             | PSNR $\uparrow$ | SSIM $\uparrow$ | LPIPS $\downarrow$ |
>   | :---------: | :-------------: | :-------------: | :----------------: |
>   |    3DGS     |      25.79      |     0.8775      |       0.2059       |
>   |    2DGS     |      25.67      |     0.8744      |       0.2265       |
>   |     GOF     |      24.76      |     0.8646      |       0.2189       |
>   |    PGSR     |      25.73      |     0.8761      |       0.2250       |
>   | DN-Splatter |      24.81      |     0.8537      |     **0.1394**     |
>   |    Ours     |    **26.42**    |   **0.8874**    |       0.2141       |
>
>   PlanarGS obtains strong visual quality, **better than our baseline method, PGSR**, while significantly improving surface reconstruction accuracy. **Visualizations of rendered results** are available in the Supplementary Material.
>
> - **Q5:** Limited experimental benchmarks: concerns about potential cherry-picking in ScanNet++（2 scenes with white walls).
> - **A5:** We conduct further experiments on two more ScanNet++ scenes — a storage room (scene 66c98f4a9b) and a bathroom (scene 88cf747085) **with colored tiled walls**. As shown in the results below. Although highly textured scenes may reduce the advantage of our planar prior supervision, PlanarGS continues to exhibit **strong reconstruction performance**. (See Table in **A1** for NVS results.)
> Reconstruction Results on 66c98f4a9b of the Scannet++ Dataset
>
>  |             | Acc $\downarrow$ | Comp $\downarrow$ | Prec $\uparrow$ | Recall $\uparrow$ | CD $\downarrow$ | F-score $\uparrow$ | Normal C. $\uparrow$ |
>   | :---------: | :--------------: | :---------------: | :-------------: | :---------------: | :-------------: | :----------------: | :------------------: |
>   |    3DGS     |       8.49       |       12.48       |      52.58      |       56.02       |      10.49      |       54.25        |        65.40         |
>   |    2DGS     |       6.74       |       12.14       |      64.09      |       58.48       |      9.44       |       61.16        |        74.94         |
>   |     GOF     |       7.78       |       9.39        |      55.05      |       63.01       |      8.59       |       58.76        |        68.72         |
>   |    PGSR     |       6.44       |       9.50        |      65.67      |       61.58       |      7.97       |       63.57        |        80.34         |
>   | DN-Splatter |       3.56       |       2.30        |      80.60      |       92.55       |      2.93       |       86.16        |        78.00         |
>   |    Ours     |     **2.88**     |     **2.15**      |    **87.20**    |     **93.00**     |    **2.51**     |     **90.00**      |      **86.64**       |
>
> Reconstruction Results on 88cf747085 of the Scannet++ Dataset
>
> |             | Acc $\downarrow$ | Comp $\downarrow$ | Prec $\uparrow$ | Recall $\uparrow$ | CD $\downarrow$ | F-score $\uparrow$ | Normal C. $\uparrow$ |
>   | :---------: | :--------------: | :---------------: | :-------------: | :---------------: | :-------------: | :----------------: | :------------------: |
>   |    3DGS     |      10.42       |       9.35        |      32.10      |       38.50       |      9.89       |       36.01        |        66.17         |
>   |    2DGS     |       6.31       |       6.78        |      57.48      |       56.29       |      6.54       |       56.88        |        84.80         |
>   |     GOF     |       6.71       |       4.55        |      60.86      |       75.76       |      5.63       |       67.50        |        70.73         |
>   |    PGSR     |       6.56       |       6.25        |      57.76      |       60.44       |       6.4       |       59.07        |        81.58         |
>   | DN-Splatter |    **2.96**     |       1.66        |    **89.49**    |     **98.22**     |    **2.31**     |     **93.65**      |        86.85         |
>   |    Ours     |       3.48       |     **1.64**      |      86.29      |       96.49       |      2.56       |       91.11        |      **90.94**       |
>
> We have to **clarify** that our dataset selection in the paper follows the **common practice in prior indoor 3DGS methods**, such as DN-Spatter, and the overall number of scenes we use is **comparable to those in related works**.
> - **Q6:** Insufficient comparisons with other baselines.
>
> - **A6:** We report results of **SuGaR** on **TnT** from prior papers, and conduct experiments on another recent Gaussian-based reconstruction method, **GOF**, as in **A7 Table**. We also compare the reconstruction results between our method and the multi-view foundation model we use—**DUSt3R**. Across all three datasets, our **high-fidelity** 3DGS pipeline significantly **outperforms** the pretrained model.
>
>
>       Reconstruction Comparison with DUSt3R on the MushRoom dataset
>
>
>   |          | Acc $\downarrow$ | Comp $\downarrow$ | Prec $\uparrow$ | Recall $\uparrow$ | CD $\downarrow$ | F-score $\uparrow$ | Normal C. $\uparrow$ |
>   | :------: | :--------------: | :---------------: | :-------------: | :---------------: | :-------------: | :----------------: | :------------------: |
>   |  DUSt3R  |       8.01       |       6.22        |      54.25      |       61.03       |      7.11       |       57.31        |        72.96         |
>   | PlanarGS |     **3.95**     |     **5.02**      |    **79.89**    |     **74.65**     |    **4.49**     |     **77.14**      |      **83.35**       |
>
>
>       Reconstruction Comparison with DUSt3R on the Replica dataset
>
>
>   |          | Acc $\downarrow$ | Comp $\downarrow$ | Prec $\uparrow$ | Recall $\uparrow$ | CD $\downarrow$ | F-score $\uparrow$ | Normal C. $\uparrow$ |
>   | :------: | :--------------: | :---------------: | :-------------: | :---------------: | :-------------: | :----------------: | :------------------: |
>   |  DUSt3R  |       7.52       |       7.18        |      41.06      |       49.84       |      7.35       |       44.89        |        68.19         |
>   | PlanarGS |     **2.80**     |     **5.45**      |    **89.28**    |     **75.67**     |    **4.13**     |     **81.90**      |      **89.88**       |
>
>
>       Reconstruction Comparison with DUSt3R on the Scannet++ dataset
>
>
>   |          | Acc $\downarrow$ | Comp $\downarrow$ | Prec $\uparrow$ | Recall $\uparrow$ | CD $\downarrow$ | F-score $\uparrow$ | Normal C. $\uparrow$ |
>   | :------: | :--------------: | :---------------: | :-------------: | :---------------: | :-------------: | :----------------: | :------------------: |
>   |  DUSt3R  |       9.70       |       6.64        |      31.08      |       50.12       |      8.17       |       38.17        |        75.27         |
>   | PlanarGS |     **3.86**     |     **3.46**      |    **81.56**    |     **84.05**     |    **3.66**     |     **82.78**      |      **90.52**       |
>
>   Actually, as in the Experiments part (line 268-271), our baselines have included the **state-of-the-art Gaussian surface reconstruction methods**, specifically **2DGS**, **PGSR**, as well as the original 3DGS and DN-Splatter that used to be **SOTA on indoor scenes** and incorporates **pre-trained models as well**. We did not include **SuGaR** in our main experiments, as it is an earlier Gaussian-based method whose performance **has been surpassed** by subsequent approaches  — including **2DGS** and **PGSR**, which we have already evaluated.
>
> - **Q7:** The proposed pipeline is not easily extendable to more diverse indoor environments, let alone outdoor scenes.
>
> - **A7:** We have additionally conducted experiments on the Meetingroom and Barn scenes from the **TnT** dataset. Meetingroom features a **large-scale, semi-open indoor space with a complex roof structure**, while Barn represents an **outdoor man-made scene**. In both cases, our method achieves **high-quality** reconstructions. ( "-" indicates that the method **failed** to generate a mesh.)
>   |             | 3DGS  | SuGaR | 2DGS  | GOF  | PGSR | DN-Splatter |   Ours   |
>   | :---------: | :---: | :---: | :---: | :--: | :--: | :---------: | :------: |
>   |    Barn     | 0.13  | 0.14  | 0.36  | 0.51 | 0.66 |      -      | **0.70** |
>   | Meetingroom | 0.01  | 0.15  | 0.16  | 0.28 | 0.29 |      -      | **0.55** |
>   |    time     | 14.3m |  2h   | 34.2m |  2h  | 1.2h |    1.1h     |    1h    |

---

> > ### Author Response · Authors · 2025-08-01
> >
> > Due to the word limit in the rebuttal phase, we address additional clarifications about your concerns here to help further demonstrate the strengths and contributions of our work. Some additional experimental results have been included in our responses to other reviewers.
> > - **Q2:** Why does the paper choose to estimate surface normals using a vision-language model combined with DUSt3R, instead of using existing indoor surface normal estimation models?
> > - **A2:** We would like to **clarify** that the core contribution (lines 75-76 in the paper) of PlanarGS **does not** lie in simply constraining 3DGS with normal priors from pre-trained models, as done in methods like DN-Splatter. Our approach instead leverages semantic priors about planar structures from **the vision-language model** to provide geometric constraints in more **holistic scales**, as discussed in the Related Work part of the paper (lines 115–119). The **coplanar constraint** is formulated in Eq10–11 in the Methods section of the paper.
> > Our co-planarity constraint has effectively enforced surface smoothness over large planar regions, as shown in the ablation study visualizations in our Supplementary Material **(Fig. 5)**, and as explained in the Experiments of our paper (lines 278–281 and 285–286), as well as **Tables 1–4**. The **improvements are evident** compared to methods that rely solely on geometric priors from pre-trained models, such as DN-Splatter.
> > Finally, it is the **multi-view foundation model** that we leverage to provide geometric priors, which are **multi-view consistent** compared to monocular priors, as mentioned in Related Work (line 119).
> > - **Q3:** Does the method require modifying the prompt for each scene? If so, it would hinder generalization.
> > - **A3:** **No.** We use **exactly the same prompts** for all various scenes in ScanNet++ and Replica and add "sofa, blackboard" in the complex Mushroom dataset consistently across all scenes, as stated in the Implementation Details section of our paper (lines 260–262). It can be seen that the prompts **do not** need to include all planes in the scene, and prompts for objects **out of the scene** are also widely used. In practice, the inclusion or omission of several prompts will not break the pipeline, only **slightly influence the accuracy** of plane segmentation and 3D reconstruction.
> >   Thus, the prompts can be easily given by both users and large language models **without careful design**. Contrary to your concern, we intentionally support open-vocabulary prompts in our pipeline to **improve generalization**, avoiding performance degradation in diverse indoor scenes that often occurs with specifically trained plane segmentation models or existing 3DGS-based methods. **Overall**, the open-vocabulary capability enhances the robustness and scalability of PlanarGS without compromising usability.
> > - **Q4:** Heavy dependence on pretrained models and concerns about robustness compared to traditional approaches such as COLMAP.
> > - **A4:** Our method **does not rely on highly accurate outputs** from the pre-trained models. Our **cross-view fusion** and **geometric inspection** modules significantly enhance the stability and effectiveness of GroundedSAM in plane segmentation tasks, as described in Section 3.2 of the paper. Furthermore, we introduce **multiple types of masks**—including confidence-based, edge-based, and plane-region masks—to mitigate the poor performance of the multi-view foundation model in certain cases, as shown in Eq14, 15 in Section 3.4 of the paper.  The joint effect of multiple loss terms on multiple views during the training process further reduces the impact of occasional errors from the priors.
> >   Additionally, PlanarGS does not depend on any particular pre-trained model. We **replace DUSt3R** with another model **VGGT**, and **please refer to our rebuttal to the reviewer q67a in A2 for these results**.
> >   Finally, as discussed in the first paragraph of the Related Work, the challenge of reconstructing large, texture-less plane surfaces has long been **a major factor** limiting the robustness of traditional 3D reconstruction pipelines. In fact, COLMAP performs very poorly in such regions, and that is why we introduce the **Planar-guided Initialization** for points supplementary as described in Lines 192–197 of Section 3.3. At the same time, our method utilizes the semantic priors to define plane regions for constraint,  **barely introducing errors** to the strong reconstruction capabilities of the original baseline. Therefore, there is no compromise in robustness in PlanarGS.

---

### Decision · Program_Chairs · 2025-09-17

**Decision:**

Accept (poster)

**Comment:**

This paper introduces PlanarGS, which integrates planar priors derived from segmentation foundation models and VLMs with depth and normal priors obtained from multi-view foundation models into the 3DGS pipeline. The goal is to address the difficulty of reconstructing large, low-texture planar surfaces in indoor scenes, where photometric consistency alone often fails to estimate accurate geometry. By introducing coplanarity constraints and depth-normal consistency terms, the proposed method guides the optimization of Gaussians toward more faithful surfaces and achieves improvements in geometric reconstruction.

The reviewers raised concerns that the novelty is somewhat limited, since the proposed approach relies on integrating existing models such as segmentation models, VLMs, and multi-view reconstruction methods, especially regarding the proposed plane segmentation pipeline. However, in the rebuttal, the authors reported that their segmentation pipeline, which combines a VLM with a foundation model, outperforms existing plane segmentation methods. The rebuttal also provided additional comparisons, ablation studies, and evaluations on more diverse datasets, which highlighted the superior performance and robustness of the proposed approach.

Overall, the proposed pipeline is well-motivated, experimentally validated, and addresses an important challenge in indoor 3D reconstruction. The improvements over baselines are consistent and convincing, and the contribution is of clear interest to the community. I recommend acceptance.